# Sensitivity analysis of the PALM model system 6.0 in the urban environment

Michal Belda[1], Jaroslav Resler[2], Jan Geletič[2], Pavel Krč[2], Björn Maronga[3], Matthias Sühring[3], Mona Kurppa[6], Farah Kanani-Sühring[3,4], Vladimír Fuka[1], Kryštof Eben[2], Nina Benešová[5], and Mikko Auvinen[6]

[1]Department of Atmospheric Physics, Faculty of Mathematics and Physics, Charles University, Prague, Czech Republic
[2]Institute of Computer Science, Czech Academy of Sciences, Prague, Czech Republic
[3]Institute of Meteorology and Climatology, Leibniz University Hannover, Hannover, Germany
[4]Harzenergie GmbH & Co. KG, Goslar, Germany
[5]Czech Hydrometeorological Institute, Prague, Czech Republic
[6]Atmospheric Composition Research, Finnish Meteorological Institute, Helsinki, Finland

**Correspondence:** Michal Belda (michal.belda@mff.cuni.cz)

**Abstract.** Sensitivity of the PALM model 6.0 with respect to land-surface and building properties is tested in a real urban environment in the vicinity of a typical crossroad in a densely built-up residential area in Prague, Czech Republic. The turbulence resolving PALM is able to simulate the urban boundary layer flow for realistic setups. Beside an accurate representation of the relevant physical processes, the model performance depends also on the input data describing the urban setup, namely the building and land-surface properties. Two types of scenarios are employed. First are the synthetic scenarios altering mainly surface and material parameters such as albedo, emissivity or wall conductivity, testing sensitivity of the model simulations to potentially erroneous input data. Second, urbanistic type scenarios are analyzed, in which commonly considered urban heat island mitigation measures such as greening of the streets or changing surface materials are applied in order to assess the limits of the effects of a particular type of scenario. For the synthetic scenarios, surface parameters used in radiation balance equations are found to be the most sensitive overall followed by the volumetric heat capacity and thermal conductivity of walls. Other parameters show limited average effect, however, some can still be significant in some parts of the day, such as surface roughness in the morning hours. Second type, the urbanistic scenarios, show urban vegetation to be the most effective measure, especially when considering both physical and biophysical temperature indicators. Influence of both type scenarios was also tested for air quality, specifically $PM_{2.5}$ dispersion which generally shows behaviour opposite to thermal indicators, i.e., improved thermal comfort brings deterioration of $PM_{2.5}$ concentrations.

# 1 Introduction

Investigation of the urban climate, and especially that of the urban heat island (UHI) phenomenon, still faces new challenges, despite decades of intensive research (Oke, 1982; Arnfield, 2003; Souch and Grimmond, 2006; Mills, 2014). Even with increasing computing capabilities and geographic information systems (GIS), there is a need for standardized research methods. Furthermore, research output should be applicable in practise (Stewart, 2011; Mills, 2014). Microscale meteorological and climate models have been increasingly used for simulations of real urban city environments and especially the impacts of changes in the city structure on the environmental conditions that affect the inhabitants. For a long time, cities have been known to strongly modify the surface energy balance and atmospheric conditions by trapping the energy in the city causing the UHI (Oke, 1982). In addition to that, global changes of climate, especially global temperature increase, are expected to have a worldwide influence on human society and other natural ecosystems with potential severe impacts (IPCC, 2014a).

The increase of heat load in urban areas has been reported to have a substantially harmful effect on public health (Patz et al., 2005; Haines et al., 2006; Ebi, 2011) with an increase of mortality rates (Kovats and Hajat, 2008; Zanobetti et al., 2012). On the other hand, when appropriate adaptation measures are applied, these negative consequences can be mitigated (Gill et al., 2007; Hunt and Watkiss, 2011; Müller et al, 2013; IPCC, 2014b). In this context, various UHI mitigation measures are being considered, with greening of the environment as a typical example. Application of these measures, however, needs some prior information about their potential effectiveness. For that, it is important to know how sensitive the environment is to the city layout (e.g., building height or street width) and the material-specific parameters used to describe urban surfaces (e.g., reflectivity or roughness).

As the public and the administrative authorities are becoming aware of the problem, so grows the demand for scientifically-based urban climate studies, particularly model-based studies that can provide reliable projections on city or street-level scale. Beside an accurate representation of the relevant physical processes in urban climate models, their performance also depends on the accuracy of the input data that define the urban environment, for example the building heights and building-physical properties, the location of trees, their shape and leaf-area density, land-surface parameters, etc. However, many model or physical parameters describing the city environment are only known approximately or are not available at all. Therefore, it is important to know the sensitivity of the model results to the uncertainties in the input data, in order to assess the spread of potential deviations in model simulations or, in planning stages, which parameters are to be gathered with higher priority in data collection campaigns.

In practice, different model types are being used for urban studies, ranging from radiation models (SOLWEIG: Lindberg et al., 2008, 2018; RayMan: Matzarakis et al., 2010), atmospheric kilometre-scale NWP/climate models with integrated urban parameterizations to detailed street-scale models. Considering their respective approach and resolutions, different model groups can give quite different answers to the potential users. Regional climate models, for example, typically use idealized street canyon schemes (e.g. Single Layer Urban Canopy Model, SLUCM: Kusaka et al., 2001; Building Effect Parameterization, BEP: Martilli et al., 2002; Building Energy Model, BEM: Salamanca et al., 2010) which can be useful for simulations of city quarters or entire cities, but given their relatively low resolution, can do so on long time scales and for large regions or even

continents. On the other side of the spectrum are very high-resolution metre-scale models that can give quite a detailed picture of individual streets and buildings, but due to computational requirements are usually limited in their spatial and temporal coverage. Our study uses the latter approach, so we limit the following state-of-the-art summary to the street-scale models.

Parameter sensitivity studies for urban flow models based on computational fluid dynamics (CFD) are rare and typically deal with parameters such as grid size/resolution or the type of turbulence model included (e.g., Ai et al., 2014, Ramponi and Blocken, 2012, Crank et al., 2018). More common are studies that consider the effect of potential changes in urban development, such as tree planting, green roofs or changes of certain surface materials, typically increasing reflectivity. For example, Ashie and Kono (2010) evaluate the impact of a redevelopment plan in two districts of Tokyo using a RANS-based (Reynolds-averaged Navier-Stokes) CFD model and Gross (2012) considers the effects of various green design elements, such as green facades, green roofs, lawns and trees also using a RANS-based CFD code. Many previous studies have also applied the RANS code called ENVI-met, though the focus has been on a small number of specific changes, instead of a systematic model sensitivity study (e.g., Su et al., 2014; Emmanuel and Loconsole, 2015; Lobaccaro and Acero, 2015). For an extensive review of available studies with description of the ENVI-met model we refer to Gál and Kantor (2020), for a comprehensive metastudy comparing methodologies and results of microscale and mesoscale models, please see Krayenhoff et al. (2021).

Large-eddy simulation (LES) is a branch of CFD in which the large turbulent eddies are explicitly resolved and simulated, unlike RANS where all turbulent eddies are parameterized. The LES method has been shown to perform better in resolving instantaneous turbulence structures in a complex urban environment (e.g., García-Sánchez et al., 2018; Salim et al., 2011; Gousseau et al., 2011; Tominaga and Stathopoulos, 2011). However, to the best of our knowledge, comprehensive sensitivity studies on how LES results for urban environments depend on the input data accuracy are non-existent to date.

This paper presents a systematic sensitivity analysis of the LES-based PALM model system 6.0 (Maronga et al., 2015, 2020) during a heat-wave period. The selected area of interest is based in a real urban district in Prague, Czech Republic. Our interest concentrates on the sensitivity of the air temperature, surface temperature, and $PM_{2.5}$ (particulate matter less than 2.5 μm in aerodynamic diameter) concentration to the parameters describing the properties of the urban surfaces. The purpose of this study is two-fold. First, to evaluate potential errors of model simulations introduced by erroneous setting of material parameters in the model (e.g., if the parameters are not measured correctly, with enough detail, or only roughly estimated). Second, to show a potential and limits of various idealized measures typically considered for urban heat island mitigation.

The paper is organized as follows: Sect. 2 describes the LES model, the numerical setup, and gives an overview of the sensitivity simulations. The results of the sensitivity analysis and mitigation measures are presented in Sect. 3. A summary and discussion of the results is given in Sect. 4.

## 2 Experiment setup

### 2.1 Model description

The PALM model system 6.0 (revision 4093) (Maronga et al., 2015, 2020) consists of the PALM model core, several embedded modules, and PALM-4U (short for PALM for urban applications) components which have been specifically developed

for modelling the urban environment. PALM model core resolves the non-hydrostatic, filtered, incompressible Navier-Stokes equations for wind $(u, v, w)$ and scalar quantities (potential temperature, water vapor mixing ratio, passive scalar) on a staggered Cartesian grid in Boussinesq-approximated form. The sub-grid scale terms that arise from filtering are parameterized using a 1.5-order closure by Deardorff (1980), with modifications after Moeng and Wyngaard (1988) and Saiki et al. (2000). One of the assets of PALM is its excellent scalability for massively parallel computer architectures (up to 50,000 processor cores, see Maronga et al., 2015).

This study applies several modules embedded in PALM, namely the land surface (LSM; Gehrke et al., 2020), plant canopy (PCM) and radiation model. The radiation model applies the Rapid Radiation Transfer Model for Global Models (RRTMG), which has been used as an external library. Furthermore, the following PALM-4U components are applied: the Cartesian topography, building surface model (BSM, formerly USM, see Resler et al., 2017), model of radiation interaction with surfaces and plant canopy so called radiative transfer model (RTM, see Krč et al., 2021), and human biometeorology (BIO, see Frölich and Matzarakis, 2020 and Krč et al., 2021) and online chemistry modules (CHEM, see Khan et al., 2021).

Additionally, both self- and offline nesting features of PALM-4U are utilised. In self-nesting a domain with a finer resolution can be defined inside a larger domain and this subdomain (child domain) receives its boundary conditions from the coarse-resolution parent domain at every model timestep (Hellsten et al., 2020). In offline nesting, the initial and boundary conditions for the mean flow of the parent domain are provided from, e.g., a mesoscale model using a dynamic driver, while the child domain receives all information from its parent (Kadasch et al., 2020). As offline nesting is usually used for coupling to a large-scale or mesoscale model that does not resolve turbulence, it is triggered at the model boundaries using a synthetic turbulence generator (STG), which imposes spatially and temporally correlated perturbations every time-step onto the velocity components at the lateral boundaries.

Two modelling domains were connected with the one-way online nesting feature of PALM (see Sect. 2.3 for more details). The initial and boundary conditions of the parent domain were taken from a WRF model simulation using the offline nesting feature of PALM-4U; the boundary conditions were updated at every model time step (Sect. 2.2.2). The WRF data were processed by the PALM supplementary WRF_interface, for description see Resler at al. (2020).

For overview of the PALM model, embedded modules and the PALM-4U components, see Maronga et al. (2020) and for details the companion papers in this special issue.

## 2.2 PALM model set-up

### 2.2.1 General model configuration

The dynamic core of the PALM model was configured with the Wicker and Skamarock 5[th] order advection scheme (Wicker and Skamarock, 2002) and the multigrid pressure solver (Hackbusch, 1985; Maronga et al., 2015). The radiative fluxes were simulated by RRTMG and their interactions with the urban canopy layer were modelled by RTM (Krč et al., 2021). The surface energy balance for the individual surfaces (vegetation, pavement, buildings, water) was calculated by the LSM and BSM components (Maronga et al., 2020). The dynamic and energy processes caused by resolved trees and shrubs were modelled

by PCM. The chemistry module was configured for $NO_X$, $PM_{10}$ and $PM_{2.5}$ species without chemical reactions and boundary conditions set to zero to simulate purely the passive transport of the emitted pollutants and consequently to simplify attribution of the sensitivity tests to local features.

To initialise temperatures of walls, grounds, and roofs, a 48-hour spin-up simulation for the BSM and LSM was conducted. During this spin-up run, the model solves only simplified energy processes while the effects of the airflow on the energy balance were held constant (see Maronga et al., 2020). The simplifications also include a simple radiation model instead of RRTMG and switching off the window treatment in BSM. The spin-up allows to establish reasonable initial temperatures inside the ground, wall, and roof material layers while keeping the computational demands within an acceptable range.

### 2.2.2    WRF model configuration

Initial and boundary conditions for the parent domain of the PALM-4U simulations were obtained from a WRF model simulation initialized from the GFS operational analyses and forecasts. WRF (version 3.8.1) was run on two nested domains with horizontal resolution of 9 and 3 km, and 49 vertical levels. The dimensions of the inner domain were 187×121 grid points. The configuration was standard: NOAH LSM, RRTMG radiation and Yonsei University scheme for the planetary boundary layer
(PBL). According to preliminary tests no urban parameterization has been used in the WRF model and the settings arising from the MODIS land use categories have not been altered. We used four runs of GFS daily, starting at synoptic times, namely 18 UTC previous day, 00, 06 and 12 UTC on the day of the simulation. From each of these GFS runs, first 12 hours were taken and downscaled by WRF. The forecast horizons 0-6 h served as a spin-up and were discarded. The remaining horizons 7-12 from each run were assembled into 24 hourly outputs per day. Thus a surrogate for local analysis has arisen, aiming at
elimination of a possible drift of WRF model fields from reality, while adding local effects not simulated by the global GFS.

     WRF outputs from the 3 km domain were postprocessed into the PALM dynamic driver. The data were transformed between coordinate systems and a horizontal and vertical interpolation was applied including terrain matching procedures. The interpolated airflow was adjusted to enforce mass conservation. The tool for processing the WRF data into the PALM dynamic driver file is a part of the official PALM distribution as WRF_interface since revision 4766, the description of this process is given in
Resler at al. (2020).

### 2.2.3    Surface and material parameters

For solving the energy balance equations, BSM and LSM require using detailed and precise input parameters describing the surface materials (e.g., albedo, emissivity, roughness length, thermal conductivity, capacity of the skin layer, thermal capacity and volumetric thermal conductivity). Urban and land surfaces and materials become very heterogeneous in a real urban
environment when going to very fine spatial resolution. Any bulk parameterization for the whole domain would be inadequate. For our study, a very detailed setting of the parameters was supplied everywhere possible. In order to obtain the data, an extensive on-site campaign was performed which provided a detailed database of geospatial data including information on wall, ground, and roof materials and colours for the estimating surface and material properties (Resler et al., 2017). The

original geo-database was extended with information about neighbouring streets and updated with new modifications (see section 2.3 for detailed description).

Surfaces are described by their respective material category and albedo. Parameters other than albedo are estimated and assigned to each category based on surface and subsurface material composition and thickness. The parameters of all subsurface layers of the respective material were set to the same value. The skin layer heat capacity $C_0$ and heat conductivity between the skin layer and the first material layer $\Lambda$ (see Equations 1 and 2 in Resler et al., 2017) were inferred from the properties of the near-surface material, which may be different in the rest of the volume. Parameter settings of the categories used in this study are given in supplements as Table S01. Trees in the analyzed domain were described by their respective position, diameter, trunk parameters and vertically stratified leaf area density. Prague 3D model available from the Prague Institute of Planning and Development was used to obtain the building height database. The Prague 3D model is based on photogrammetric (aerial) mapping and is freely available on Prague OpenData portal (https://www.geoportalpraha.cz/cs/data/metadata/44EE8B0A-641A-45E8-8DC9-CF209ED00897 - only available in Czech). Data are provided in CAD (DWG or DGN) or ESRI (polygon or multipatch) format. Original product accuracy in 2012 was 0.5m, but model is yearly updated and current accuracy is around 0.2m. Description and properties of surfaces and materials were assembled into standard GIS formats and subsequently transformed into the PALM input NetCDF files corresponding to the PALM Input Data Standard (PIDS: Heldens et al., 2020).

## 2.3 Study domain description

The study domain in Prague-Holešovice was adapted from Resler et al. (2017), covering the vicinity of a crossroad of Dělnická street and Komunardů street in a densely built-up area in Prague, Czech Republic (50°06.195' N, 14°27.000' E). The area is well suited for this type of study as it represents a typical Prague residential area in a rather topographically flat (terrain elevation ∼180 m a.s.l.) part of the city with a variety of urban components, including old and new residential buildings, backyards and parking spaces. The two streets run north to south (Komunardů) and west to east (Dělnická), and have the width of roughly 25 and 17 m, respectively. The buildings in the area range approximately from 10 to 35 m in height. There is not much vegetation in the area and the majority of the trees is located in the courtyards. The surrounding neighbourhood is very similar to the study area (Fig. 1 right).

A few minor modifications were made to the study domain from the previous analysis of Resler et al. (2017). Firstly, the horizontal extent of the domain was extended from the original 376 m × 226 m to 400 m × 256 m. This was important for the domain multiplication in a synthetic domain setup (see section 2.4); the new domain ends in the middle of streets in all directions. Secondly, the central part of the intersection, where a small asphalt polygon (∼11 m$^2$) in the real street was partially replaced by cobblestones (∼7 m$^2$ of cobblestones and ∼4 m$^2$ of asphalt), was modified in the input data accordingly. Last minor change from the previous analysis is the height of the highest building which was physically rebuilt and is now 35 m high. The domain covers the area of 102,400 m$^2$, of which 48,356 m$^2$ is the total building footprint, 48,356 m$^2$ (∼22.9% of total domain surface area) are impervious surfaces and 5,593 m$^2$ (∼2.7%) are pervious surfaces (e.g., grass). Each building has three levels – lower, often markets and shops; upper, typically residential; and roof. Lower level is covered by 9,933 m$^2$ (∼4.7%)

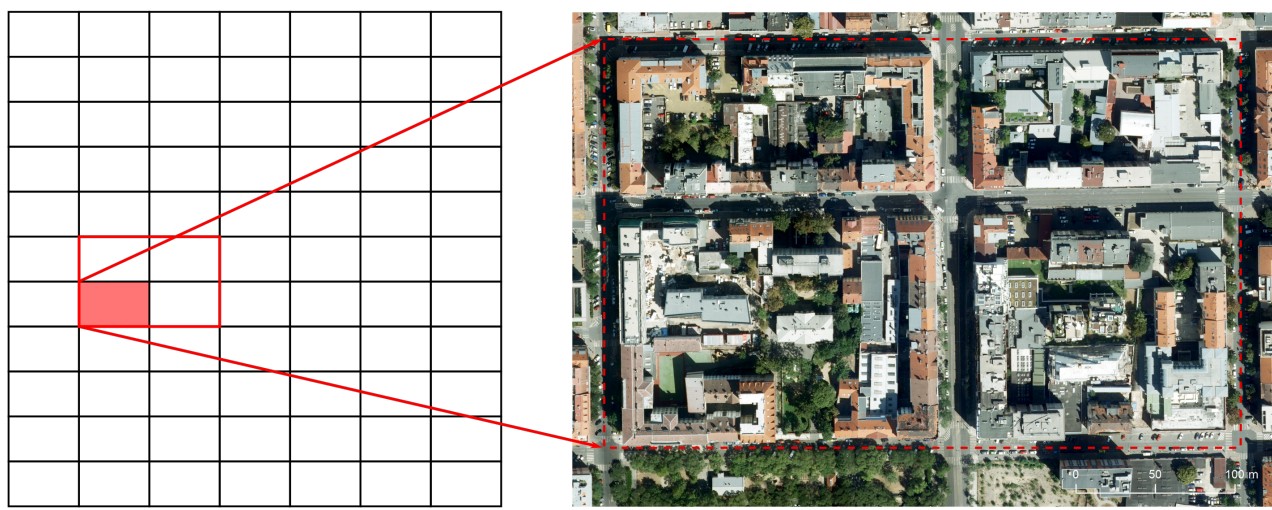

**Figure 1.** Design of model domains; black-bordered rectangles representing the parent domain, red-bordered rectangle representing the child domain. Solid red rectangle represents one unique domain with the real environment before multiplication. Projection: WGS84/UTM zone 33N, orthophoto source: Institute of Prague Development.

of windows and 20,837 m$^2$ ($\sim$9.9%) of walls, upper level is covered by 22,861 m$^2$ ($\sim$10.8%) of windows and 52,169 m$^2$ ($\sim$24.7%) of walls. Roof area is 51,044 m$^2$ ($\sim$24.2%). Total area of all surfaces in the domain is 210,793 m$^2$. In the time of
this study, 158 trees are planted in the area of which 4 are coniferous and 154 are broad-leaved.

## 2.4   Synthetic modelling domains

The study domain described above is too small for realistic large-eddy simulations, because the largest turbulent eddies are of size of the boundary layer height, which in Europe can reach up to 2.5 km in summertime (e.g., Seidel et al., 2012 or Zhang et al., 2013). In order to resolve the turbulent transport of these eddies, the horizontal model domain size must be at
least 2–3 times the boundary layer height and thus be in the order of several square kilometres, which is much larger than the employed model domain in the present study (Resler et al., 2017). Moreover, to allow simulations of real meteorological conditions, non-cyclic boundary conditions with offline nesting were considered, using the meteorological model WRF and a synthetic turbulence generator. This setting, however, requires a sufficient horizontal extent of the domain to allow development of the correct turbulent flow. For this purpose, a nested two domain setup with one-way online nesting was utilized as described
in section 2.1 and synthetic domains were generated by horizontal multiplying of the original domain.

The parent domain had a horizontal grid spacing of 8 m and was created by seven repetitions of the original domain in west-east direction and eleven repetitions in south-north direction. Moreover, an additional flat buffer zone was added on all sides of the domain. The width of this buffer was 25 grid cells at the west and east boundaries and 24 grid cells at the south and

north boundaries. Thus, the extent of the complete parent domain is 400×400 grid cells (3200 m × 3200 m) in both directions. The domain was configured with 120 vertical layers using the layer stretching approach so that the vertical grid spacing of 8 m was stretched above 120 m by a factor of 1.08 until a grid spacing of 24 m was reached. The resulting domain top was at 2.5 km.

The nested fine resolution domain (hereafter child domain) was configured with a refinement ratio of 4, having a 2m grid resolution in all directions and it consisted of four original domains; two in west-east direction and two in south-north direction. The extent of the domain was 400×256×40 grid cells (800 m × 512 m × 80 m). The child domain was located asymmetrically in the left part of the parent domain and the evaluation was done on the south-west part of it (see Fig. 1). This configuration was selected due to an easterly wind flow during the modelled episode.

## 2.5 The modelled heatwave episode

This study focuses on modelling the thermal comfort and therefore a heatwave episode on 2–3 July 2015 was chosen for these simulations. One advantage of this choice is that the previous version of the model was also validated on this period (see Resler et al., 2017). A detailed description of the weather during the modelled period is also provided in Resler et al. (2017). The weather was characterized by a high-pressure system centred above the Baltic Sea with mostly clear skies and the daily maximum temperature exceeding 30 °C while the minimum not falling below 20 °C (tropical night). Relative humidity values ranged from 30% during the day to 65% at night. Easterly winds were observed with values mostly below 2.5 m.s$^{-1}$ above the roof level. Maximum wind speed of 3–4 m.s$^{-1}$ in 10 m height was observed at the Karlov station (WMO 11519; around 4km south from the modeled domain) in the afternoon of 2 July 2015, during the spin-up, and at the end of 3 July 2015. According to the atmospheric sounding, a low-level jet from the south and south-east was observed during the night, with a maximum wind speed of 10 m.s$^{-1}$ at 640 m a.s.l. (950 hPa). At night, a south/south-east low-level jet was observed in the atmospheric soundings, with a 10 m.s$^{-1}$ maximum wind speed at 640 m a.s.l. (950 hPa). The time of the sunset was 19:15 UTC on 2 July 2015, sunrise at 2:58 UTC and solar noon at 11:06 UTC on 3 July 2015.

## 2.6 Air pollution and emissions

Air pollution sources in the modelling domain are dominated by the local road traffic. Based on the Czech national emission database, the mobile sources represent approximately 60% of total emissions for Prague for particulate matter and 75% for NO$_x$ (CHMI, 2018). Considering those ratios and the fact that there is no major point source in the area, we decided to include only the traffic sources to the analysis. The emission fluxes are estimated based on the daily traffic intensities, which are available from annual traffic census data, for all streets in both directions. Emission factors, taken from a local Czech database (MEFA, 2013), give pollutant release per vehicle per meter of travel, based on vehicle and fuel type. For our study area, the assumption was that all vehicles are passenger cars, which is reasonable for this residential neighbourhood. The traffic-related emissions are spatially uniformly distributed into traffic lanes and temporally distributed using prescribed hourly factors also derived from available annual traffic census data (see Fig. S10 for daily spatial distribution). Magnitudes of emission fluxes range from 0.03 to 0.34 g·day$^{-1}$·m$^{-2}$ for NO$_x$, from 6 to 58·10$^{-3}$ g·day$^{-1}$·m$^{-2}$ for PM$_{10}$ and from 3 to 32·10$^{-3}$ g·day$^{-1}$·m$^{-2}$ for PM$_{2.5}$. We chose

the $PM_{2.5}$ to be the pollutant of interest, however, considering the emission creation methodology and the fact that all chemical reactions are omitted in our simulations, the conclusions (in a qualitative sense) would be the same for other pollutants. We opted not to include interactive chemistry and only consider the dispersion of $PM_{2.5}$ due to the time frame of the secondary aerosol formation being considerably longer than the lifetime of air in the domain and thus not significantly influencing the sensitivity experiments (see Sect. 4.2 for discussion of this limitation).

## 2.7  Sensitivity tests

For evaluating the influence of the parameter changes, a *baseline* simulation was performed in which the parameters tested were set to "real" values, that is, values measured or estimated based on materials used in the actual buildings and other surfaces in the domain. The scenario simulations, divided into two groups, synthetic and urbanistic, then changed one or more of these parameters as described in the following two sections.

### 2.7.1  Synthetic scenarios - sensitivity to the setting of material parameters

For the first group of sensitivity tests, a suite of synthetic scenarios was selected based on the most important variables in the urban environment. These scenarios target potential biases in the model outputs connected to the imprecise setting of relevant city environment parameters which have a major influence on the energy balance and dynamics of the model such as albedo or roughness. These parameters are notoriously difficult to obtain with a sufficient resolution and are thus usually set in a very general way and sometimes even tuned to the model results. As model errors can stem from many different sources, such as model deficiencies, chaotic behaviour or imperfect input data, we aim to quantify which part of the error can be attributed to the setting of these parameters.

Since the analysis by Resler et al. (2017), the PALM modelling system has been extended with new features. According to the new functionalities, window and wall fractions were mapped for each building in BSM and more detailed plant canopy parameters were included in PCM. In total 21 scenarios (hereafter "SA" scenarios) were prepared that each changes one specific parameter of the surfaces (and/or plant canopy) from the baseline simulation. Table 1 summarizes the parameter changes for the SA scenarios, the surfaces affected by the change and the fraction of the total surface area affected in the respective scenario.

### 2.7.2  Urbanistic scenarios - sensitivity to urban heat island mitigation measures

The second group of scenarios was designed more from the urban planners' point of view, i.e., assessing the influence of (in)appropriate urban planning actions to improve thermal comfort and air quality. These scenarios present several measures typically taken into account when dealing with the UHI effect, such as greening or changes in the surface materials, simplified to distinguish individual influence (e.g., when changing roads to grass the emissions are not changed). Not necessarily realistic, these scenarios provide the urban planners with an assessment of maximum potential influence of certain common types of urban development (e.g. removal of all trees versus planting trees everywhere). The design of the scenarios stemmed from the discussion with various authorities of the City of Prague in the framework of the Urbi Pragensi project

**Table 1.** Scenarios testing model sensitivity to changes of material parameters with fraction of affected domain surface area (column Surf. fraction). Detailed description of surfaces is in section 2.3.

| Scenario | Description | Surfaces | Surf. fraction (%) |
|---|---|---|---|
| SA01 | Albedo increase +20% | Walls, roofs, surfaces | 100.0 |
| SA02 | Albedo decrease -20% | Walls, roofs, surfaces | 100.0 |
| SA03 | Emissivity set to the average for each group of surfaces | Land cover: 0.8922; Lower walls: 0.9263; Upper walls: 0.9278; Roofs: 0.7233 | 100.0 |
| SA04 | Average SA03 emissivity +20% | Average = SA03, max. 1.0 | 100.0 |
| SA05 | Average SA03 emissivity -20% | Average = SA03 | 100.0 |
| SA06 | Roughness increase +20% | Walls, roofs, surfaces | 100.0 |
| SA07 | Roughness decrease -20% | Walls, roofs, surfaces | 100.0 |
| SA08 | Thickness increase +20% | Walls, roofs, surfaces | 100.0 |
| SA09 | Thickness decrease -20% | Walls, roofs, surfaces | 100.0 |
| SA10 | Transmissivity of windows increase +20% | Walls (windows only) | 15.6 |
| SA11 | Transmissivity of windows decrease -20% | Walls (windows only) | 15.6 |
| SA12 | Thermal conductivity inside of wall increase +20% | Walls | 34.6 |
| SA13 | Thermal conductivity inside of wall decrease -20% | Walls | 34.6 |
| SA14 | Volumetric heat capacity increase +20% | Walls, roofs, surfaces | 100.0 |
| SA15 | Volumetric heat capacity decrease -20% | Walls, roofs, surfaces | 100.0 |
| SA16 | Window fraction increase +20% | Walls | 18.7 |
| SA17 | Window fraction decrease -20% | Walls | 12.5 |
| SA18 | Leaf area density increase +20% | Trees | |
| SA19 | Leaf area density decrease -20% | Trees | |
| SA20 | Soil moisture increase +20% | Pervious surfaces only | 2.7 |
| SA21 | Soil moisture decrease -20% | Pervious surfaces only | 2.7 |

**Table 2.** Scenarios testing sensitivity of the model results to UHI mitigation measures

| Scenario | Description | Note |
|---|---|---|
| SB01 | Building height increase +20% | Street canyon ratio |
| SB02 | Building height decrease -20% | Street canyon ratio |
| SB03 | All surfaces (pavement) changed to asphalt | Land cover |
| SB04 | All surfaces (pavement) changed to concrete | Land cover |
| SB05 | All surfaces (pavement) changed to cobblestones | Land cover |
| SB06 | All surfaces (pavement) changed to white cobblestones | Land cover |
| SB07 | Tram green line | Land cover |
| *SB08\** | *All surfaces insulated* | *Walls only* |
| SB09 | Water channel instead of tram line, roads were changed to grass | Land cover, no changes in emissions |
| SB10 | Green areas changed to asphalt, trees were deleted | Grey city 1 |
| SB11 | Asphalt except main roads and pavements changed to grass, all trees deleted | Grey city 2 |
| SB12 | Planted trees on each possible place; placed 128 acer platanoides | Green city |
| SB13 | New tree alley: Dělnická, center-line position | Acer platanoides |
| SB14 | New tree alley: Dělnická, both-side position | Acer platanoides |
| SB15 | New tree alley: both streets, both-side position | Acer platanoides |
| SB16 | All trees coniferous | More dense crown |
| SB17 | Include anthropogenic heat flux | A/Cs, heating etc. |

\* Scenario SB08 was removed from further analysis, because results were significantly affected by numerical instability solved in PALM SVN revision 4240.

(http://www.urbipragensi.cz). Detailed description of this group of scenarios (hereafter denoted by the "SB" prefix) is included in Table 2.

**3 Results**

Due to a different nature of the two sets of scenarios, the analysis of the model results will be performed separately for the synthetic SA scenarios and urbanistic SB scenarios. However, some aspects of the analysis are common for both. Chaotic nature of the turbulent flow in the domain requires an application of time averaging which needs to be sufficiently long to smooth out turbulent fluctuations, yet short enough to capture the diurnal variability. In the time series plots, we opted to show 270 10-minute averaged values together with hourly moving averages. Summary tables (Table S02 in the supplement), on the other hand, show three-hour averages along with daily averages, minima and maxima. One important aspect of the modelling setup which must be kept in mind when analyzing the results is that the model spin-up period uses a constant dynamic and simplified

energy model (see section 2.2.1) and thus the initial thermal conditions (grounds, walls, and roofs temperatures) are not in total agreement with temperatures that would have been obtained by a full model run. This can impose differences of the simulation behaviour in the first hours from standard behaviour in the following hours when this initial effect vanishes, which may limit the applicability of the results in the first few hours of the simulation after the spin up. However, as most of the differences between respective simulations begin to appear after sunrise, this influence can be neglected.

Spatial variability is analysed by averaging over the whole domain as well as separately over several selected domain parts. A particular focus is on the two crossing streets and courtyards. For maps with point positions and area selections, see the supplemental material (Fig. S01-S09). The most important variables for the end users were chosen as primary indicators. They include surface temperature, air temperature, $PM_{2.5}$ concentrations and two biophysical temperature characteristics: mean radiant temperature (MRT) and physiological equivalent temperature (PET) all in the height of the human body represented by the first 2m high layer (for definitions and information about the implementation of MRT and PET in PALM see Frölich and Matzarakis, 2020 and Krč et al., 2021).

All scenario simulations are analyzed with respect to the baseline simulation (a model run with the original parameter values). Figure 2 shows spatial distribution of basic variables in the domain for the baseline simulation.

## 3.1 Sensitivity to material parameters

In the first part of the assessment, we analyze the model sensitivity to the setting of building and material parameters such as albedo or roughness (SA scenarios). Figure 3 shows the sensitivity as differences between respective scenarios and the baseline simulation for air temperature (other variables are included in the supplement as Fig. S11-S14) in 24-hour averages. Table S02 in supplementary files summarizes all five analyzed variables showing absolute values and differences (plus relative difference) of each SA scenario from the baseline. Results are averaged for several areas: domain, east-west street (Dělnická), south-north street (Komunardů), both streets (Streets) and courtyards. In general, the following four parameters show the highest sensitivity in temperature in the following settings: albedo (SA01, SA02), emissivity (SA03–05), thermal conductivity of walls (SA12, SA13) and volumetric heat capacity (SA14, SA15) with median response up to $\pm0.1$ K (Fig. 3) and maximum response reaching up to $\pm0.18$ K in three-hour averages and up to $\pm0.4$ K in 10-minute averages for some parameters during the day (Table S02). Overall, the albedo setting (SA01, SA02) shows the highest sensitivity of all parameters in this group. The lowest sensitivity is observed for wall thickness (SA08, SA09), transmissivity of windows (SA10, SA11) and soil moisture (SA20, SA21). However, the reason for the low sensitivity to the changes of the soil moisture lies mainly in a low percentage of the green areas in the domain.

The daily cycle of air temperature also has an imprint in the relative importance of respective parameters throughout the day. Parameters used in incoming radiation routines (namely albedo; SA01, SA02) are the most sensitive ones in the middle of the day, when the radiative balance is governed mostly by incoming shortwave radiation. During the night, emissivity (SA03–05) and heat capacity of walls (SA14, SA15) play a major role (see Table S02), thus sensitivity to these parameters is higher then. Some parameters show quite high sensitivity only in short periods during the day. For example window fraction shows low sensitivity in the morning hours, after which it increases at around 9–12 UTC (11–14 local time) and peaks in the early evening

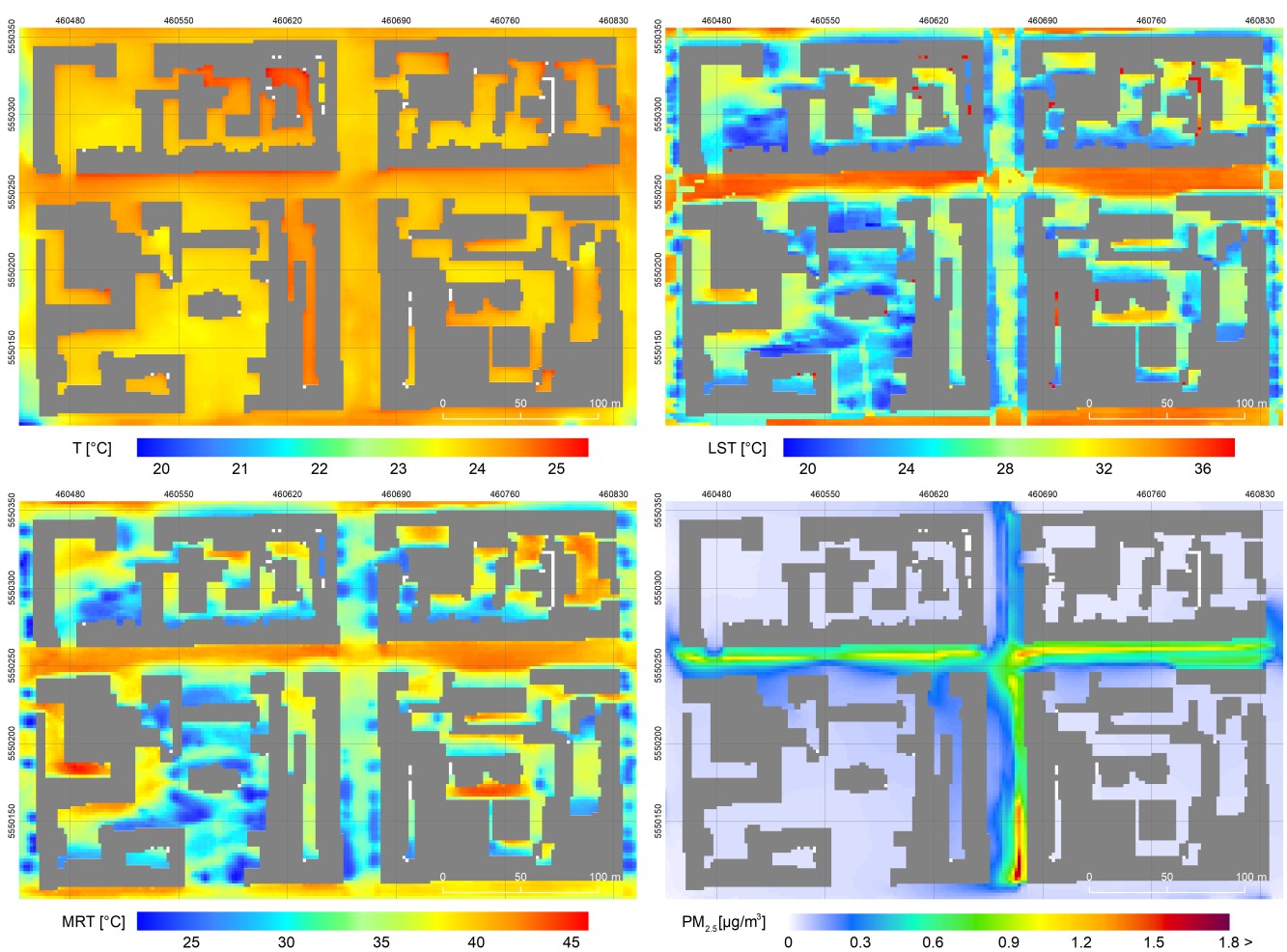

**Figure 2.** Daily average spatial variability of air temperature (top left), surface temperature (top right), mean radiant temperature – MRT (bottom left) and PM₂.₅ concentrations (bottom right) for the baseline simulation. Projection: WGS 84/UTM zone 33N; layer with roofs is own data source.

at around 18–21 UTC (see SA16–17 in Table S02). In this particular case, given that the response to lower window fraction is an increase of temperatures and vice versa, the most likely explanation is the difference in heat storage between windows (very low) and walls (higher), which has a prevalent influence in low-sun periods of the day.

Air temperature, though fundamental for physical evaluation, is not necessarily the best quantity for evaluating biophysical properties, namely thermal comfort. For this purpose, MRT and PET variables combining other relevant physical variables (radiation, humidity, air flow, etc) are used. Given the combination of various influences, MRT and PET often show higher but inverse sensitivity than the air temperature. As a demonstration we show this on the two most prominent scenarios SA01 and SA02. In SA01, the albedo is increased by 20%, which results in the decrease of daily surface temperatures by 0.5 K and a

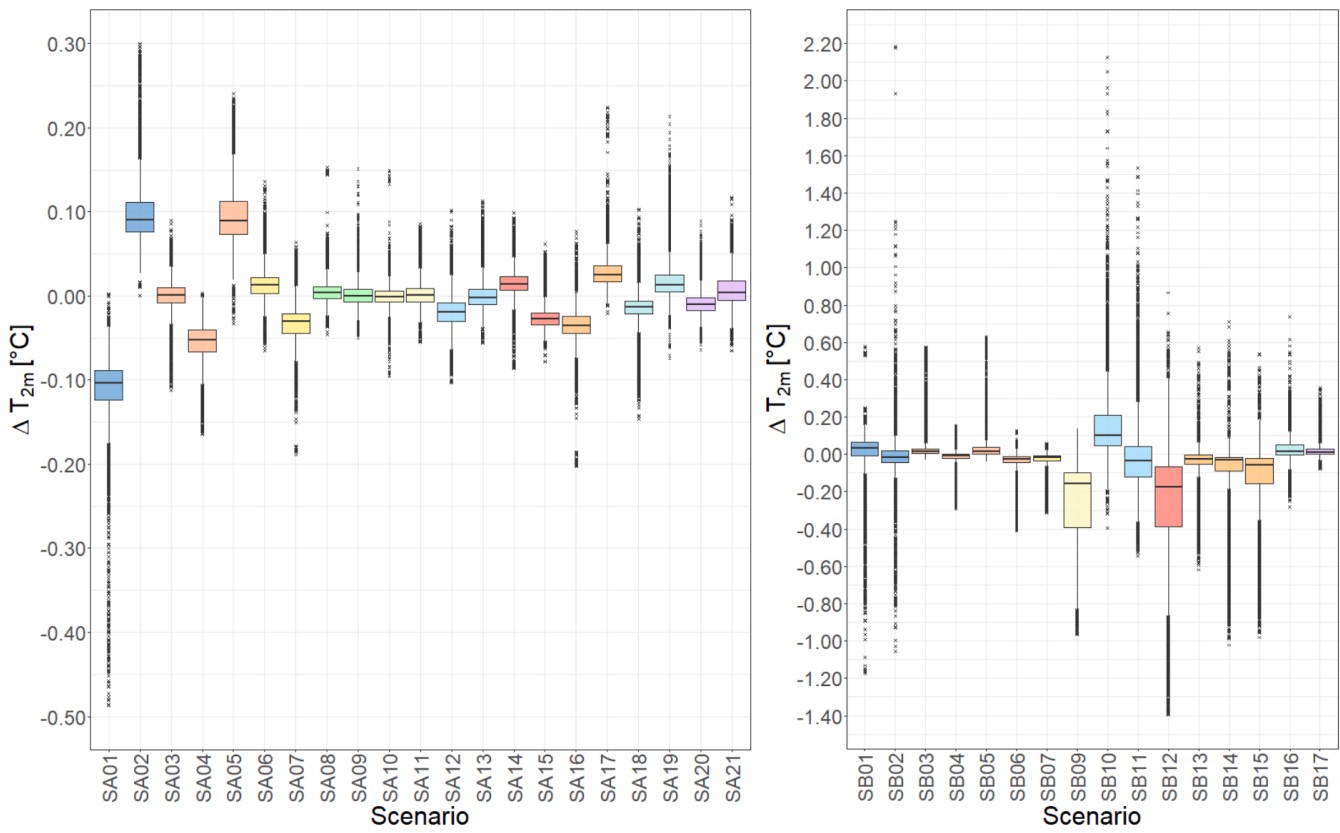

**Figure 3.** Sensitivity of air temperature in SA (left) and SB (right) scenarios. Values represent gridbox differences (scenario-baseline) of 24-hour averages in the first 2m high layer. Box colors indicate related scenarios (e.g., blue: changing albedo, orange: changing emissivity, etc.). Whiskers: values within 1.5x interquantile range; crosses: outliers.

decrease of around 0.1 K for air temperature. On the other hand, by increasing reflection at the surfaces, this change increases both MRT and PET by 0.6 K and 0.3 K respectively. In daily maxima, the increase of both biometeorological variables is even more prominent and reaches up to 1.7 K and 1.6 K respectively. Decreasing albedo by 20% in SA02 has a similar effect in absolute numbers with the opposite sign.

    Influence on air quality, represented here by changes of $PM_{2.5}$ concentrations at the first model layer, originating from
emission from local transportation, is generally much less pronounced in all scenarios. For the dominant parameters, such as albedo or emissivity, we still observe a similar general tendency to increase (decrease) $PM_{2.5}$ values with increased (decreased) albedo (emissivity). This behaviour is opposite to the surface and air temperatures and it is likely primarily caused by connected changes in the flow regime as illustrated in Fig. 4 and 5 by decrease (increase) of wind speed with increased (decreased) albedo (also discussed in e.g. Žák et al., 2016). It should be noted here, that due to non-linearity, the response to the symmetrically
constructed scenarios (e.g. SA01 and SA02) need not be symmetric in the spatial distribution as also illustrated in Fig. 4 and 5.

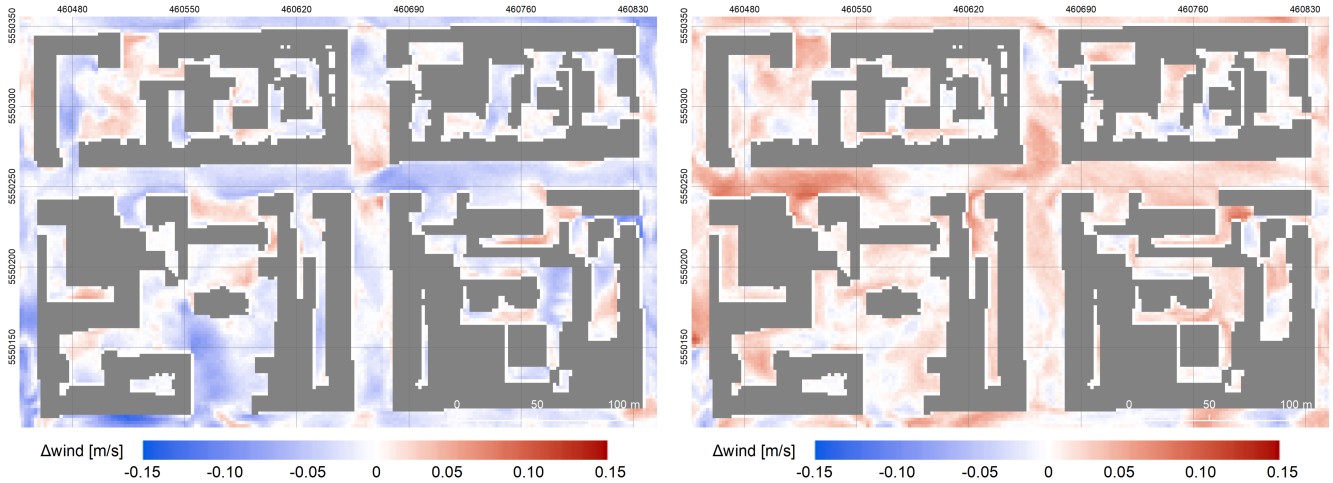

**Figure 4.** Daily average sensitivity of horizontal wind speed (1m) expressed as the difference between scenario and baseline. Left: scenario SA01 (albedo increased by 20%); right: scenario SA02 (albedo decreased by 20%). Projection: WGS84/UTM zone 33N, layer with roofs is own data source.

For example, the changes in wind speed are more pronounced in the western part of the west-east oriented street and at the crossroad when decreasing the albedo. Furthermore, the sensitivity in some places, e.g. the northern part of the north-south oriented street or some courtyards is such that decreasing or increasing albedo both result in increasing wind speed.

Long-term average changes of PM$_{2.5}$ concentrations are generally small and with the exception of singular peaks (Fig. 6) lying within $\pm5\%$ in most of the domain. The temporal evolution of the response, however, may also differ depending on the geometric configuration as is also evident from Fig. 6 which shows spatially averaged values for the two main streets in the albedo changing SA01 and SA02 scenarios. The difference between the two scenarios is more pronounced in the north-south oriented Komunardů street in the afternoon hours, while in the morning hours, the difference is larger in the west-east oriented Dělnická street.

The parameters we analyse influence the results mainly by changing the energy balance of the horizontal and vertical surfaces in the model domain. Air temperature changes are then mainly driven by the transfer of heat between these surfaces and air. In this context, we will now focus on the effect on surface temperatures. The highest sensitivity of surface temperature is observed in the same scenarios as for air temperature; albedo SA01, SA02 (Fig. 7), emissivity SA03–05, thermal conductivity SA12, SA13 and volumetric heat capacity SA14, SA15. The average response reaches up to $\pm0.5$ K and the 3-hour maxima up to $\pm0.9$ K with albedo changes (SA01, SA02) and decreased emissivity (SA05).

The model response to the surface parameters is also dependent on the location. This stems mainly from the differences in the radiation budget during the day caused by positioning of urban elements (buildings and trees). At individual points, the differences of surface temperature with respect to the base case reach up to $\pm4$ K in shorter periods in the albedo change scenarios SA01 and SA02 (e.g., points C02, C05 in the upper two panels of Fig. 7).

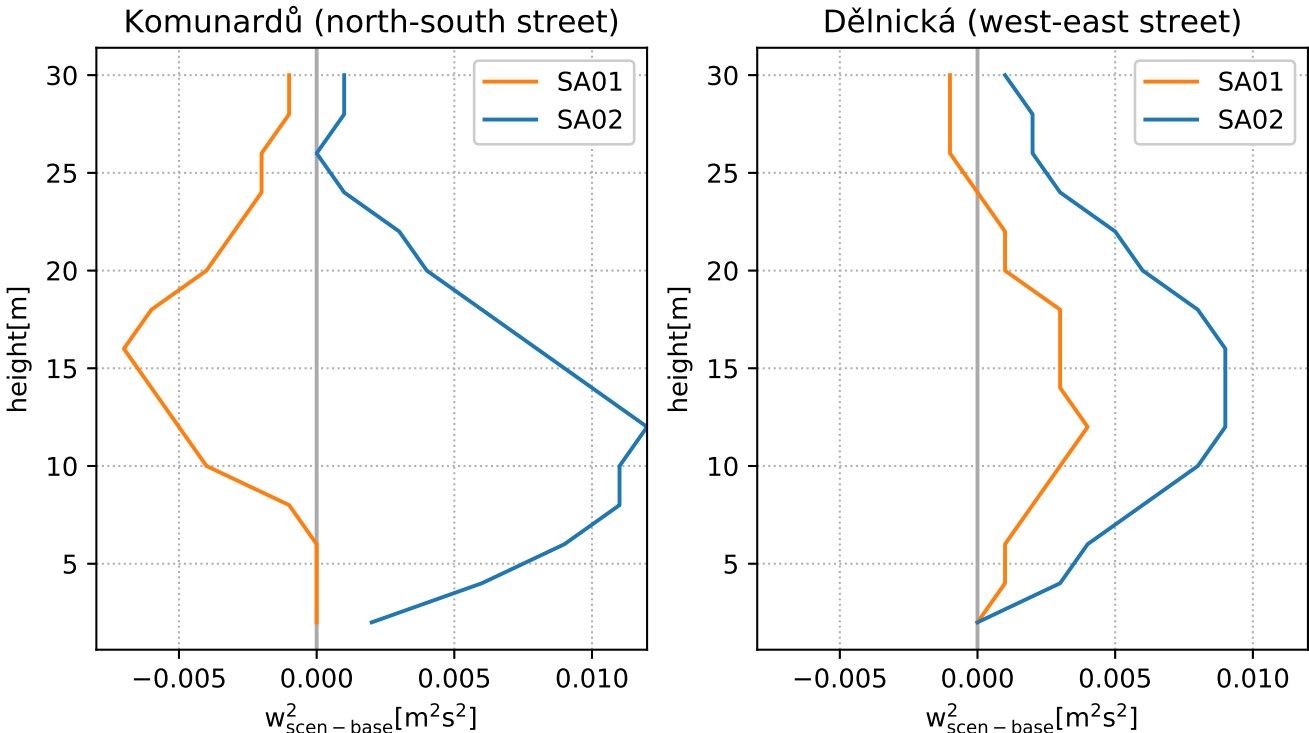

**Figure 5.** Daily average profiles of $w^2$ (plotted as the difference $scenario - basecase$) averaged over the two main streets: the north-south oriented Komunardů (left) and the west-east oriented Dělnická (right).

Air temperature showed a rather small sensitivity to changes in soil moisture, which we attribute to a rather low percentage of green areas in the domain. However, small areas covered with or in direct vicinity of vegetation are influenced significantly as shown in Fig. 8 for point F03. In this and other similar points (e.g., F02, F04, H02), changes in soil moisture show much higher sensitivity in surface temperature and biometeorological indicators (in the additional outputs, see Code and data availability section for URL) around noon with differences reaching up to 6 K. For other examples of the influence of soil moisture on surface temperature in a validation study of a real city environment see also Resler at al. (2020).

In some parts of the domain, the typical daily cycle of the differences is even reversed in certain periods of the day. A typical example of this behaviour is the sensitivity of surface temperature to albedo changes (Fig. 7). While most surfaces show an expected increase (decrease) of temperature with the decrease (increase) of albedo (typical examples are points C02, C05 in Fig. 7), some analysis points (eg. A02, A04, B04, B06, D13, D14) show reverse influence. Two examples of the inverse behaviour are illustrated in Fig. 7 for points A02 and D14 (lower two panels), clearly showing higher (lower) albedo resulting in higher (lower) surface temperatures in some parts of the day, when presumably increased (decreased) reflection from other surfaces brings more (less) SW radiation at these points compared to the base case. The difference in the incoming SW radiation for points A01 and A02 is demonstrated in Fig. S15 comparing the S01 (blue) and S02 (orange) scenarios with baseline (black):

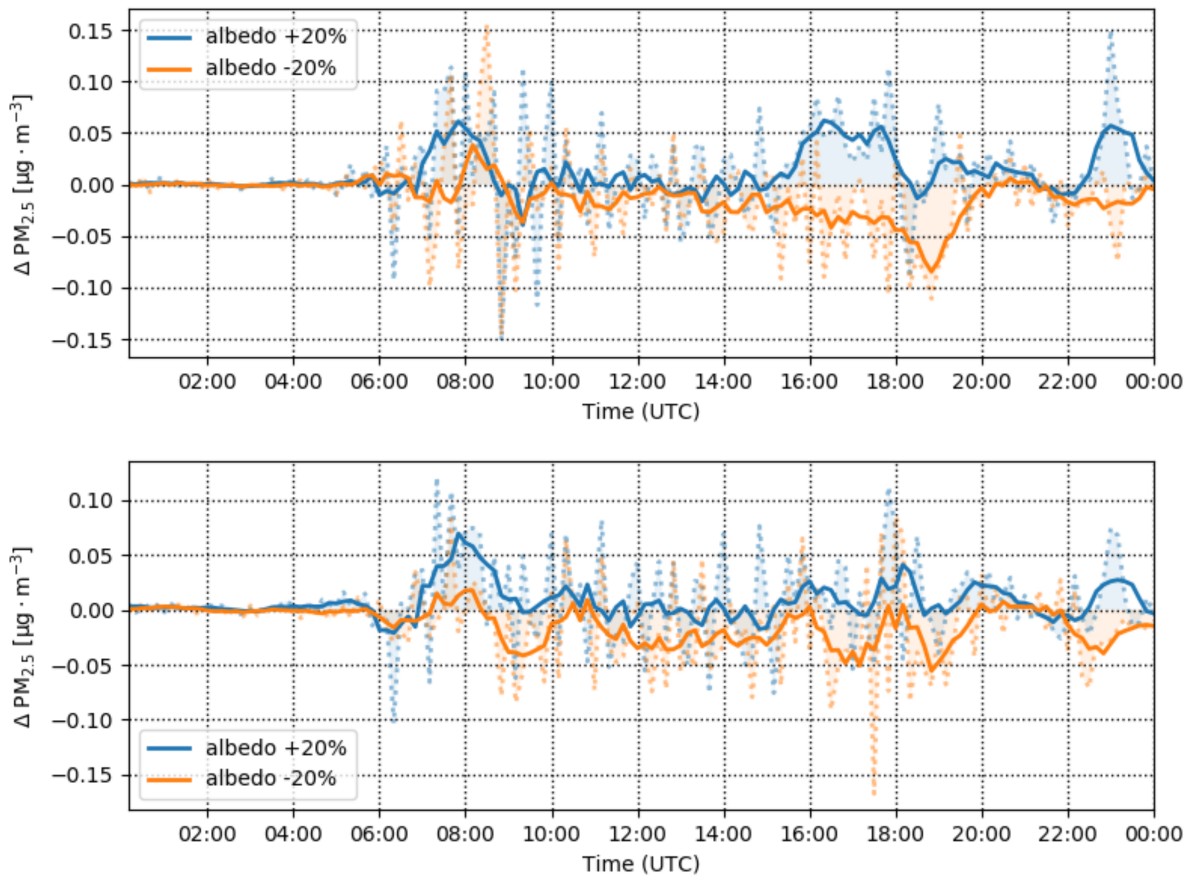

**Figure 6.** Changes of PM$_{2.5}$ for the north-south oriented Komunardů street (top) and the west-east oriented Dělnická street (bottom) in scenarios SA01 (albedo+20%, blue) and SA02 (albedo-20%, orange). Dotted: 10-minute values; solid: 1-hour moving average.

the A01 point (solid lines) receives less incoming radiation with increased albedo, while for the A02 point (dashed line) the
incoming radiation is increased with increased albedo due to reflection from opposite surfaces in the corresponding time.

High spatial variability is also evident from other scenarios and shows the importance of using very high resolution models
for local studies. As can be seen from e.g. figures for the emissivity changing scenarios SA03-SA05 (in the additional outputs,
see Code and data availability section for URL), while the spatially averaged response shows mostly a simple daily cycle with
maximum change around noon, some individual points (eg. A02, A04) show maxima in the morning and afternoon hours,
while around noon the effect diminishes. Depending on the individual surface radiation budget given by the incoming solar
radiation and reflections from other surface, some points experience a delayed peak in the afternoon hours with an inverse
response, i.e. increased emissivity leads to increase of temperatures (C04, D02).

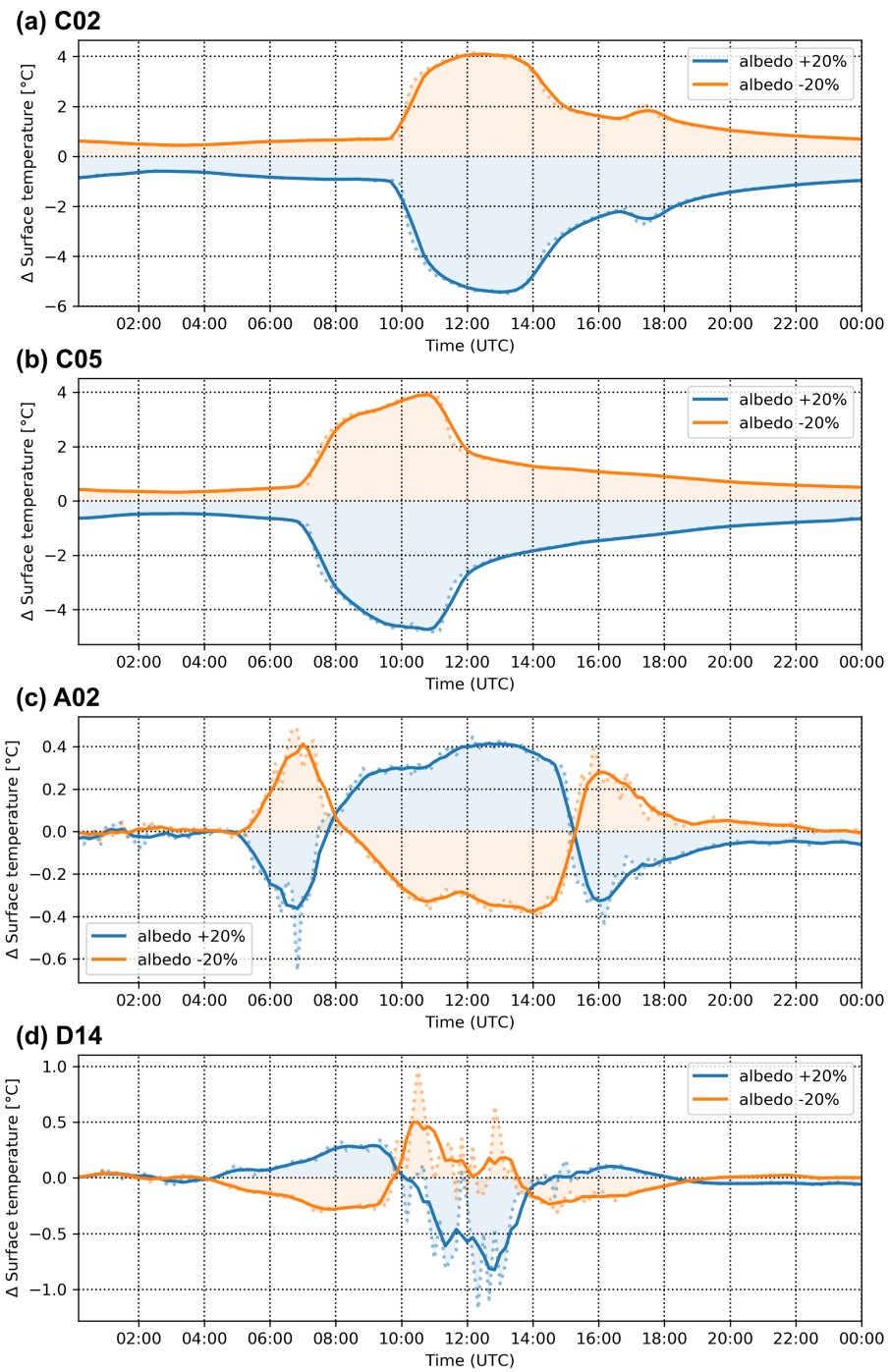

**Figure 7.** Differences in surface temperature in evaluation points C02 (a), C05 (b), A02 (c) and D14 (d) for albedo changing scenarios SA01 (blue) and SA02 (orange). Dotted: 10-minute values; solid: 1-hour moving average.

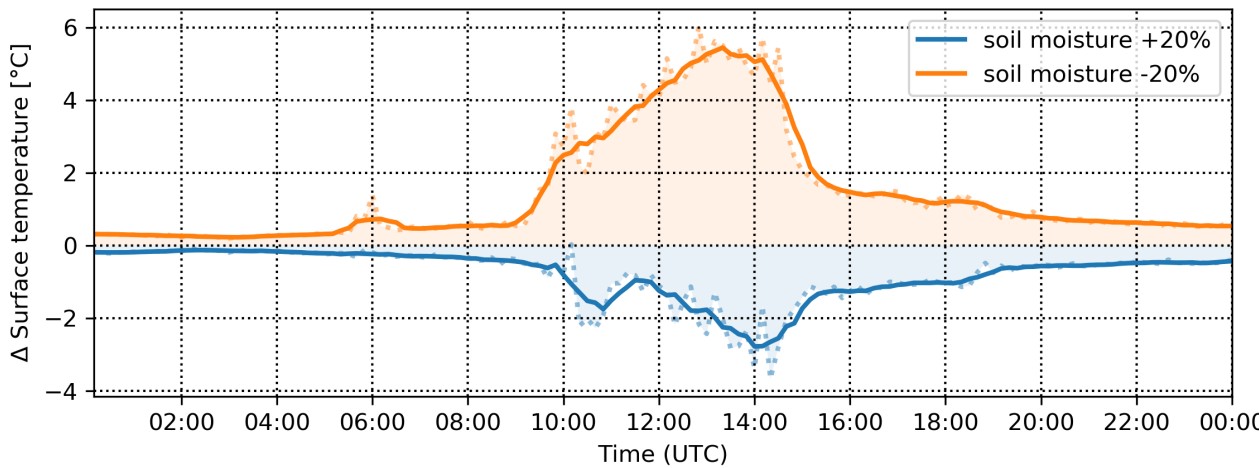

**Figure 8.** Soil moisture sensitivity of surface temperature difference for scenario SA20 (blue line) and SA21 (orange line) at point F03. Dotted: 10-minute values; solid: 1-hour moving average.

## 3.2 Sensitivity to urban heat island mitigation measures

The second part of the sensitivity analysis focuses on the urbanistic scenarios. These scenarios include UHI mitigation mea-
sures, which planners and decision makers might apply to improve the bioclimatic situation in the city during high temperature conditions, especially heatwaves. Typically considered measures include planting trees or changing surface materials (Table S02). As a contrast to SA scenarios, SB scenarios usually require changing more than one parameter at once. For instance, replacing concrete with grass results in changes in albedo, emissivity, roughness as well as other parameters.

 Sensitivity of the model response to SB scenarios is also summarized in Table S02. The most significant changes in sur-
face temperature are observed in scenarios SB09 (land cover changes), SB10 (grey city 1), SB11 (grey city 2), while for air temperature (Fig. 3), SB09–11, SB12 (green city with many planted trees), SB14 (new tree alley with both-side position on Dělnická street) and SB15 (new tree alley with both-side position on both streets) show the strongest sensitivity. Scenario SB09, in which grass replaces roads and a tram line is replaced with a water channel, shows a decrease of surface temperatures by up to 3.0 K and up to 0.3 K for air temperature. Grey city scenarios SB10 and SB11 (Fig. 9), on the other hand, tend to
increase temperatures significantly with 3-hour maximum differences exceeding 2 K on the horizontal surfaces, whereas for air temperatures an increase by 0.3 K and 0.1 K, respectively, is found. However, this difference between the two scenarios is dependent on the area of interest. For example, in the north-south street (Komunardů), the change in air temperatures is much more consistent between the scenarios with maxima reaching +0.5 K in the late afternoon (Table S02 in supplement).

 Scenario SB12 (green city with many planted trees) appears the most effective in decreasing temperature during the day
with surface temperature cooler by up to 4.0 K and air temperature by almost 0.5 K (Fig. 9). The effect is smaller during the

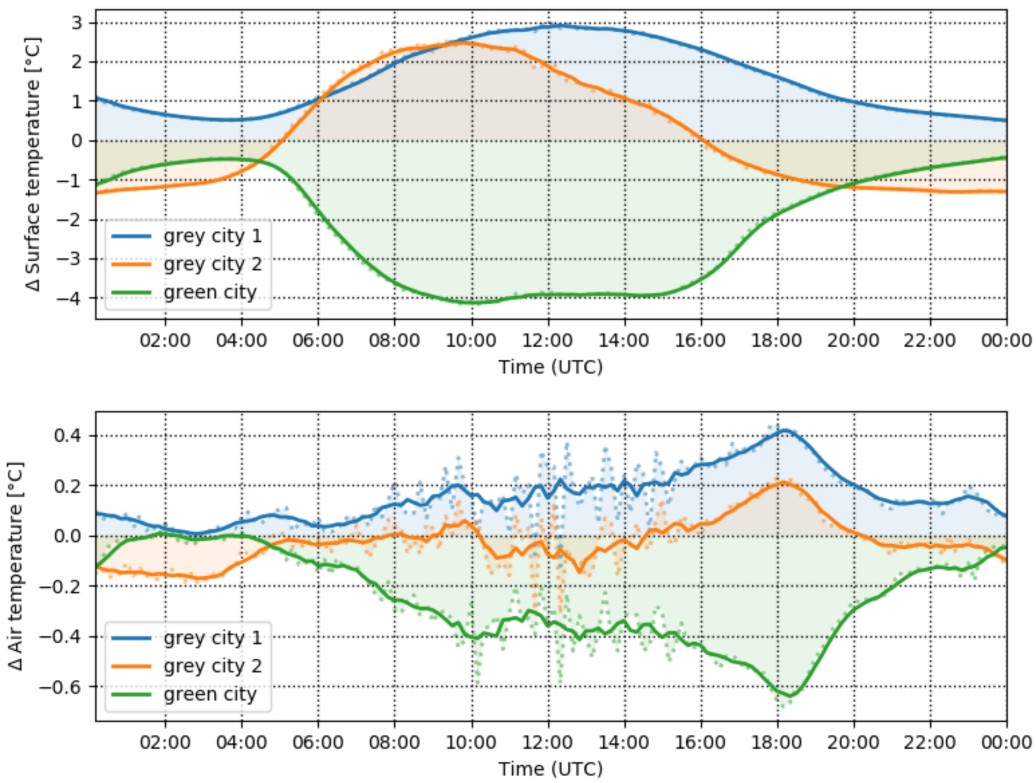

**Figure 9.** Domain-averaged differences in the surface temperature (top) and air temperature (bottom) for grey city scenarios SB10 (blue), SB11 (orange) and green city scenario SB12 (green). Dotted: 10-minute values; solid: 1-hour moving average.

nighttime, when the decrease in temperature is 0.8 K and 0.12 K, respectively. Instead, scenario SB09 and even SB11 (removing trees but increasing grass covered area) show decreases of more than 1.0 K and 0.15 K in the surface and air temperatures.

In terms of thermal comfort, the two analyzed characteristics (MRT and PET) show a behaviour qualitatively similar to the physical temperatures. Again, the SB12 scenario (green city with many planted trees) shows the most effective reduction with

maximum decrease around 9 K in MRT and 4 K in PET in the entire domain. However, the effect varies considerably in space. The strongest change is observed in the west-east oriented Dělnická street while the north-south oriented Komunardů street shows a much smaller decrease of 0.0–1.2 K (Fig. 10). This difference can be partly attributed to the geometric orientation of the streets and consequent differences of insolation during the day, but also to the actual number of trees added with respect to the base case, in which more trees already grow in Komunardů st. Similar behaviour is shown in SB13–SB15 scenarios (new

tree alley(s) scenarios) with decreases up to 4.0 K in MRT and 1.6 K in PET on average.

On the other hand, SB10 and SB11 scenarios (grey city 1 and 2) show a significant increase of both biophysical properties. The MRT is increased by 8 K (5 K) and PET by 3 K (1.6 K; see Fig. 11) around noon in the entire domain in SB10 (SB11). Similarly to the previous comparison, there is a marked spatial difference throughout the domain. However, the effect is

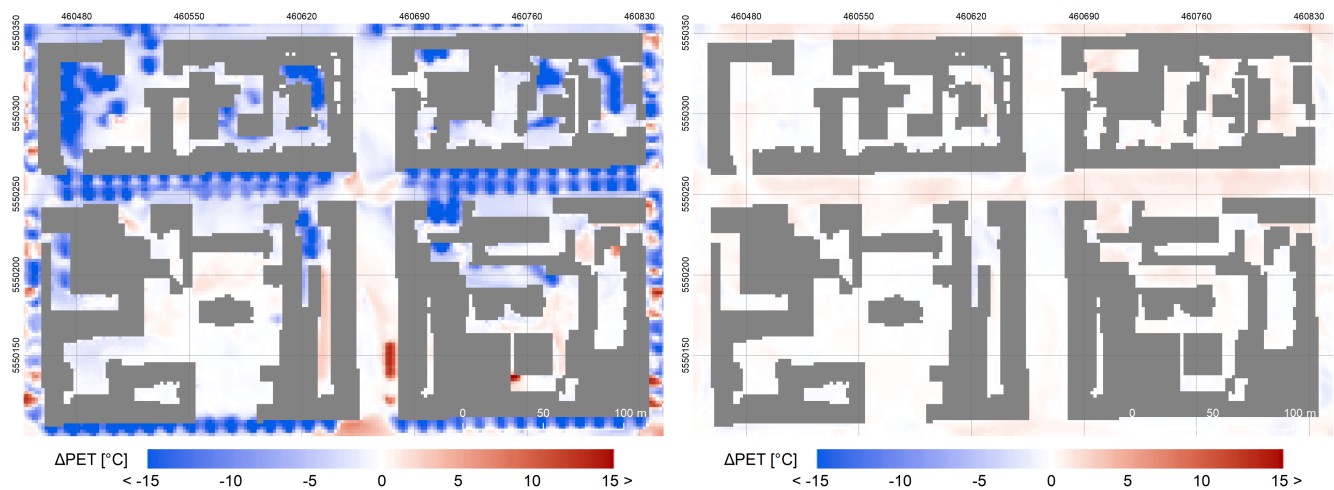

**Figure 10.** Example of spatio-temporal variability of 3-hour PET differences for a green city scenario SB12 at 09:00-12:00 UTC (left) and 21:00-24:00 UTC (right). Projection: WGS84/UTM zone 33N, layer with roofs is own data source.

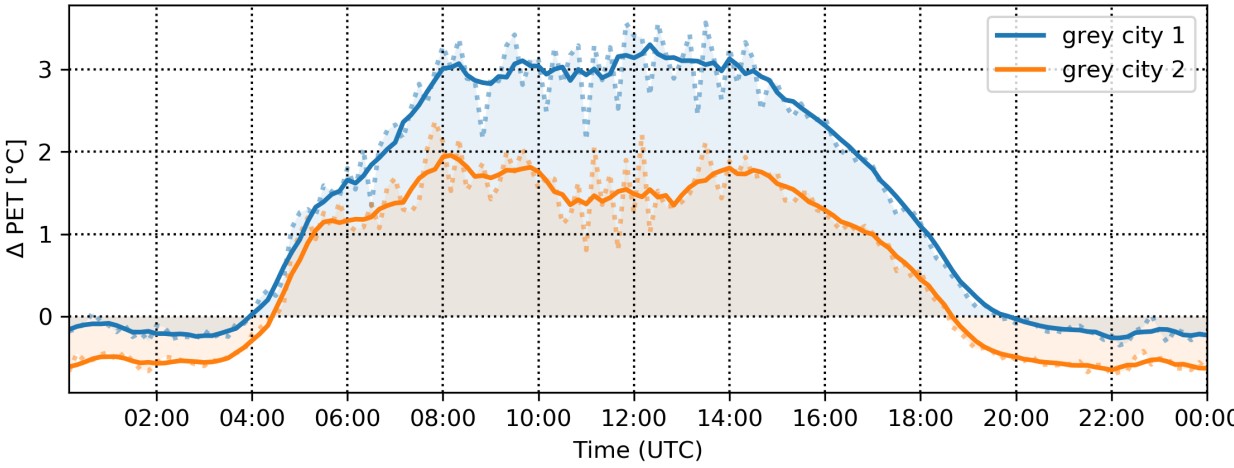

**Figure 11.** Domain-averaged differences in PET for a grey city scenarios SB10 (blue) and SB11 (orange). Dotted: 10-minute values; solid: 1-hour moving average.

strongest in the Komunardů street, with an increase of over 12 K (MRT) and 3 K (PET), and courtyards (over 9 K/4 K), while in the Dělnická street, the increase is only around 3 K in MRT and 1 K in PET.

Unlike for the sensitivity cases SA, $PM_{2.5}$ shows a significant dependence on the measures applied. However, the influence is almost universally inverse to the one for temperature. Generally, decreasing surface/air temperature increases $PM_{2.5}$ concentrations by suppressing ventilation and turbulent mixing. On average the strongest effect is observed in SB12 (green city with

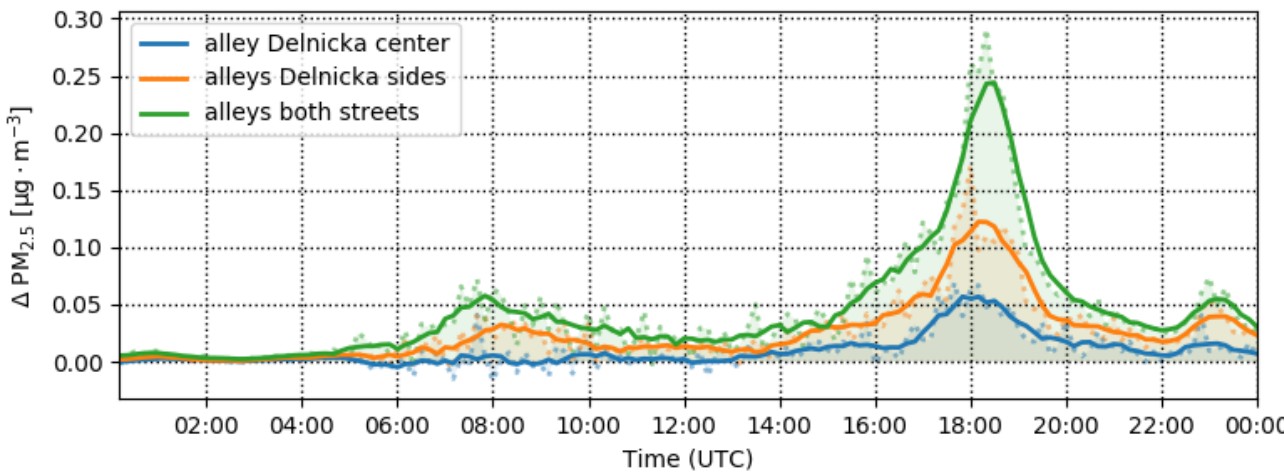

**Figure 12.** Domain-averaged differences in PM$_{2.5}$ for a new tree alley scenarios SB13 (blue), SB14 (orange) and SB15 (green). Dotted: 10-minute values; solid: 1-hour moving average.

many planted trees) and SB15 (planting the most amount of trees) scenarios (Fig. 12), which show an increase of 24% and
21% in PM$_{2.5}$ with maxima over 30% in the late afternoon hours. Scenarios that simulate planting trees only in the Dělnická street, SB13 one tree alley in the center) and SB14 (tree alleys on both sides of the street), show similar responses in terms of the shape of the daily cycle, but with a lower overall increase (Fig. 12): on average, these scenarios show an increase of 5–14% in PM$_{2.5}$ concentrations, with maxima reaching almost 20% for SB14 and 10% for SB13 scenario (Fig. 12).

Interestingly, over the perpendicular Komunardů street with no new trees planted, the concentrations tend to decrease
throughout most of the day, although this decrease is mostly concentrated in the crossroad; for the street sections north and south further away from the crossroad no significant changes are modelled. The effect is connected to spatial changes and intensification of the street canyon eddy induced by the tree-obstructed Dělnická street which acts effectively as a part of the street canyon (not shown). The grey city scenarios SB10 and SB11 conversely show decreased PM$_{2.5}$ concentrations of around 20% in the afternoon and evening. Considering the spatial differences, the highest decrease is observed in the Komunardů
street (over 50%; see Fig. 13).

## 4   Discussion and conclusions

### 4.1   Discussion

In this work, we assessed the sensitivity of air and surface temperature, MRT, PET and PM$_{2.5}$ within the PALM model system 6.0 as a response to the modification of basic surface material parameters as well as to common UHI mitigation strategies. For
this we performed a set of semi-idealized model simulations for a diurnal cycle in a city quarter in Prague.

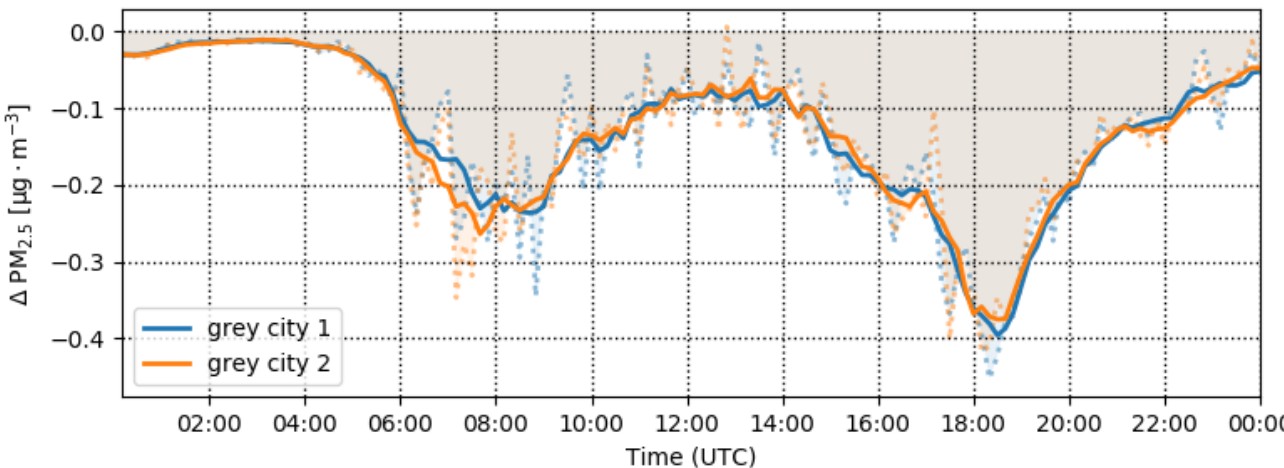

**Figure 13.** Differences in PM$_{2.5}$ in Komunardů street for a grey city scenarios SB10 (blue) and SB11 (orange). Dotted: 10-minute values; solid: 1-hour moving average.

The first set of scenarios, designed to examine the sensitivity to the parameter settings, shows the importance of the correct setting of the radiation parameters albedo and emissivity. This can be expected as the solar radiation is the main source of energy in the surface energy budget. Additionally, unlike some other parameters, radiation parameters are changed for all surfaces.

In addition to albedo and emissivity, thermal conductivity of walls and volumetric heat capacity of the materials play an important role. Other parameters show a limited average effect on the diurnal timescale, which, however, can be quite significant in some parts of the day, such as surface roughness in the morning hours and window fraction in the evening. Changing soil moisture by 20% is shown to be negligible overall in the context of the chosen domain, with only a small percentage of the surface covered by vegetation (see Table 1) except for surface temperature in the high-sun part of the day. Individual parts of

the domain with larger coverage of vegetation show greater influence. Note that we investigated only the short-term response of the urban canopy on the outlined modifications. The trends might be more prominent if long-term storage of energy in the materials was considered, i.e., when simulating a full heat wave.

   The second part of the sensitivity analysis focused on the UHI mitigation measures. One of the commonly considered measures is to paint surfaces white to increase surface albedo. However, our results indicate that this is only effective for

lowering the surface and air temperature. In contrast, the biophysical indicators MRT and PET tend to be negatively affected, i.e., thermal comfort in the street is deteriorated due to increasing the amount of reflected radiation (note that the effect can be different on purely horizontal surfaces such as roofs). Improving both physical and biophysical temperature indicators requires application of other measures, such as urban greening at the same time. Similar findings have also been reported in e.g., a meta-analysis of 52 ENVI-met simulations by Tsoka et al. (2018), Yang et al. (2015) who stress the need for precaution when

adopting high-reflectivity surfaces or Aflaki et al. (2017) who found low-albedo vegetation effective for reducing mean radiant temperature.

    Urban vegetation is found to be the most effective measure when considering reduction of both physical and biophysical temperature indicators. Conversely, grey city scenarios that reduce the amount of urban vegetation show significant worsening of the thermal comfort. Urban greenery is very often found an effective mitigation tool for UHI, for example a recent study by
McRae et al. (2020) reports vegetation-induced cooling of more than 3°C in an ENVI-met simulation. However, some studies (e.g., Wang et al., 2016, Tsoka et al., 2018 or Makido et al., 2019) show that for the best effect it is necessary to combine several measures and also to consider that different parts of the city may need different measures.

    One of the most important results of our analysis is that it confirms an opposite behaviour of thermal comfort and air quality indicators (see example in Fig. 14). Observed in both types of scenarios, the $PM_{2.5}$ concentrations typically increase with
decreasing temperatures and vice versa. The main reason for this behaviour is decreased ventilation in the street canyon due to air flow blocking. The decreased vertical turbulent transport due to reduced urban canopy temperatures and thus buoyancy can play a role too, as shown by Huszár et al. (2018b, 2020) who found significant PM decreases due to urban canopy induced vertical eddy diffusion. However, in these simulations, only aerosols passive transfer was taken into account and thus the results may be different for other air quality indicators, e.g. when considering the influence of changing reaction coefficients
and decrease of solar radiation for ozone chemistry (Huszár et al., 2018b).

    The $PM_{2.5}$ concentrations in Fig. 14 also show an important added value of the high-resolution CFD models for urban modelling compared to parameterized urban schemes in NWP/climate models or radiation models. In this case it is the spatial variability within the streets. As can be seen, the combined radiative and dynamical effects manifest in quite a heterogeneous response where (in this particular case) the increase of concentrations is most prominent on the northern half of the Dělnická
street, while in some parts on the southern side the models shows decrease of concentrations. In the Komunardů street, the response shows very small changes in the upper section of the street while in the lower section the model shows almost a see-saw response with increased concentrations in the upper half and decrease in the lower half of the section.

### 4.2    Study limitations

This study applied the PALM model revision 4093. The model itself and the configuration applied for this study have some
limitations, with the following being the most important ones in our case:

    – The model is configured without the PALM-4U building energy model (BEM) and the building inner temperature is considered constant (300 K) during the simulation. The impacts of the absence of a more complex indoor model differ in summer and winter seasons. In winter, assuming that the rooms are heated to the exact prescribed temperature by either direct local heat sources or by long-distance heating with the heating plant being outside the modelled domain, the
model adds correct heat fluxes to the insides of the buildings, albeit not providing the amount of heating energy consumed among its outputs. In summer, the constant indoor temperature can be seen as a simplification for buildings without air conditioning where the wall insulation and wall heat capacity dampen most of the daily temperature difference, as long

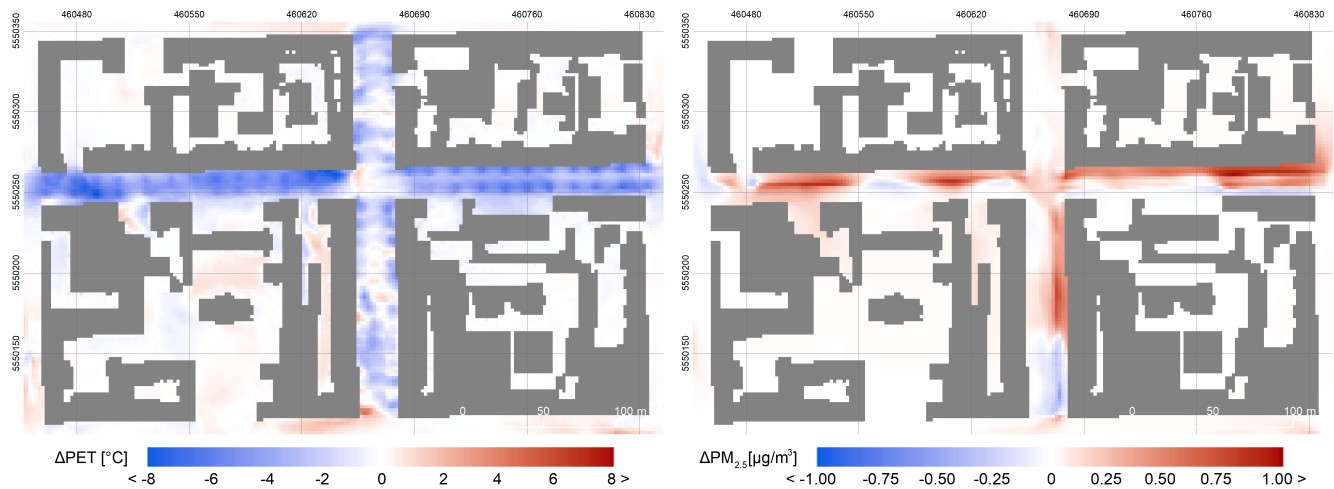

**Figure 14.** Example of opposite behaviour of thermal comfort and air quality indicators represented by average daily difference in PET and PM$_{2.5}$ for a new alley scenario SB15

as the inner temperature is realistic and the daily total net heat flux is near zero. Buildings with air conditioning need a more complex indoor model with correctly placed heat exchangers (windows for individual A/C units and roofs for central A/C systems). For the simulated domain, there was no information available about the amount and placement of A/C systems, with the majority of the buildings being old apartment houses with presumably no central A/C systems and no visible individual A/C units at windows. For long term simulations, missing waste heat which could be provided by PALM's indoor model will be important. Given the short time scale of the present study simulations, the indoor model should not affect the outcome, though. The outer wall layers react very fast to changes in the surface energy balance, but the inner wall layers have large inertia so that nothing is likely to change if the indoor temperature changes in time.

– The model sensitivities are tested only during meteorological conditions of heat wave episodes as the main focus is on simulation of the UHI mitigation measures. Only the short-term response of the urban canopy was investigated. The behaviour, including long-term response, during other seasons and weather conditions can and probably will differ from presented results.

– The simulations do not consider any chemical reactions or aerosol dynamic processes of air pollutants, only the dispersion of traffic-related PM$_{2.5}$ is considered. Moreover, the boundary conditions of the chemical species on the parent domain were set to zero. This experiment design was selected as the focus of the study is on the sensitivity of the concentrations on the local conditions. The time needed for secondary organic aerosol (SOA) formation is much longer than the typical time the chemical species spent in the studied domain (e.g., Du et al., 2018 or Tang et al., 2018). The consequence is that the SOA concentration field is almost constant over the studied domain. It means that even though

the SOA constitutes an important part of the PM$_{2.5}$, their omission does not change the differences of PM$_{2.5}$ between particular scenarios.

- This version initiates the building wall properties through the `building_2d` property in the model static driver, i.e., the wall properties are set to the roof grid cell over the wall (i.e. border grid cells of the roof). This leads to two simplifications:

    - The properties of the wall can be set only in two height zones and the corner grid cells set the properties of two surface grid cells corresponding to different walls.

    - The roof properties in the border grid cells are initialized to the wall properties. This limitation leads to artefacts in roof and wall surface temperature and heat fluxes. This drawback was removed in later versions (model revision 4240 and later) by implementation of reading separate properties for individual surface cells from the new static driver variable building_surface_pars.

- The ventilation of very tight areas surrounded by high buildings is underestimated by the model and the temperatures and concentrations of pollutants become unrealistically high in some circumstances. It is known that higher concentrations can be expected in enclosed spaces due to low turbulence (Gronemeier and Sühring, 2019). This problem was addressed in the model revision 4110. For the purpose of this analysis, these small areas were excluded from the evaluation.

Taking these limitations into account, we consider the simulation to produce plausible results both in actual values and their spatial and temporal distribution in the baseline simulation. This was confirmed by comparing the general agreement of the results to the previously validated simulations (Resler et al., 2017) in the preparatory stage of this experiment (not shown). Obviously, extensive validation of the model against observations is beyond the scope of this manuscript. For systematic validation of the current model version we refer the readers to the accompanying paper by Resler at al. (2020).

The LES simulations are quite demanding in terms of computational power, especially when compared with their RANS-based counterparts. Since we did not perform evaluation of the model results against observations in this study, it cannot show whether the additional computer resources used bring about an improvement in the model performance. However, numerous studies have been published showing the added value of LES for street canyon simulations especially when air quality is concerned. For example Gousseau et al. (2011), Salim et al. (2011) and Tominaga and Stathopoulos (2011) evaluated LES and RANS simulations against wind-tunnel measurements in a street canyon experiments and all conclude that LES shows better performance. More recently, Antoniou et al. (2017) studied outdoor ventilation in a real urban area of Nicosia, Cyprus, again evaluating RANS and LES simulations against wind-tunnel measurements. They conclude that LES simulations show smaller deviations from the measurements than RANS for mean wind speed and turbulence intensity.

## 4.3 Conclusions

In conclusion, this analysis shows that the proper setting of urban surface parameters is crucial for high-resolution LES models of the urban environment and that collecting this large amount of data is an essential part of the modelling technique. High tem-

poral and spatial variability also shows the importance of using truly local information for each area of interest. This fact also poses certain limits on the applicability of findings of this study for other locations. One the one hand, comparisons with other studies above showed qualitatively similar results in average behaviour, and in this sense we can expect similar average results in other densely built urban areas in similar climatic conditions (e.g. many European cities). However, quantitative assessment is largely dependent on the location studied, namely the physical configuration of buildings and other urban components, and thus the actual sensitivity values may differ between locations. This is evident e.g. from the albedo scenarios, for which the response in some locations was reversed depending on the geometrical configuration, or soil moisture scenarios, when the small amount of existing vegetation limits the potential response of the system to changing soil moisture in larger areas. When assessing the very local influences, e.g., pedestrian-level thermal comfort, the local settings play a major role and thus need to be considered for proper evaluation.

Altogether, the LES method proves to be an asset thanks to its capability to fully resolve the flow and to consider heterogeneity in the modelling domain. Hence, LES modelling results can be really applied to support urban planning when aiming to mitigate UHI in urban neighbourhoods.

*Code and data availability.* The PALM model system is freely available from http://palm-model.org (last access: 30 March 2020) and distributed under the GNU General Public Licence v3 (http://www.gnu.org/copyleft/gpl.html) , last access: 30 March 2020). The model source code of version 6.0 in revision r4093, used in this article is also available via https://doi.org/10.25835/0068421 (Geletic et al., 2020).

Model configuration files, input data needed for running the simulations, model outputs postprocessing code, i.e. extraction and visualisation scripts, together with necessary data extracted from the raw model outputs, and additional outputs are available for download at http://hdl.handle.net/11104/0309669.

*Author contributions.* MB was the main coordinator of manuscript proceedings and responsible for the general topic of paper and the analysis of results. All co-authors contributed to the manuscript text. KE configured and processed WRF simulations used for preparation of boundary conditions, JG was involved in geodata preprocessing, result postprocessing and data mining. JR and PK were strongly involved in PALM model setup and processed the PALM simulations, JR also participated in the experiment design. VF, FKS, MS and BM participated with the general topic, discussion and text preparation. NB, MA and MK are specialists in air quality modelling and participated in this field of study.

*Competing interests.* The authors declare that they have no conflict of interest.

*Acknowledgements.* The simulations were performed on the HPC infrastructure of the Institute of Computer Science of the Czech Academy of Sciences (ICS) supported by the long-term strategic development financing of the ICS (RVO:67985807) and partly in the supercomputing centre IT4I which was supported by The Ministry of Education, Youth and Sports from the Large Infrastructures for Research, Experimental

Development and Innovations project "IT4Innovations National Supercomputing Center – LM2015070" and partly on the HPC infrastructure of Institute of Computer Science supported by the long-term strategic development financing of the Institute of Computer Science (RVO:67985807).

Financial support was provided by the Operational Program Prague – Growth Pole of the Czech Republic project "Urbanization of weather forecast, air-quality prediction and climate scenarios for Prague" (CZ.07.1.02/0.0/0.0/16_040/0000383), which is co-financed by the EU. The co-authors Björn Maronga, Farah Kanani-Sühring and Matthias Sühring were supported by the Federal German Ministry of Education and Research (BMBF) under grant 01LP1601 within the framework of Research for Sustainable Development (FONA; https://www.fona.de/de/, last access: 27 May 2021). Financial support was also provided by the Norway Grants and Technology Agency of the Czech Republic project TO01000219: "Turbulent-resolving urban modeling of air quality and thermal comfort".

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
