# Peer review of "Sensitivity analysis of the PALM model system 6.0 in the urban environment"

_Geoscientific Model Development, 2020_

## Referee Comment (RC1) · Anonymous Referee #1 · 13 Dec 2020

General comments: This manuscript presents a series of sensitivity simulations of the PALM model in a real urban area in Prague, Czech Republic. The sensitive tests were conducted two-fold by changing the model's physical properties and changing urban morphological characteristics. All the simulations were conducted for a specific hot day over a small block area, and then their results were compared in terms of the influence of UHI and air quality. This study presents a potential capability of using the PALM model as a tool for urban climate research, and it seems that intensive computational works have been done for the sensitivity experiments. However, this study has some significant drawbacks that should be improved. First, it is not clear what this study contributes because its purpose is not clearly presented in a scientific sense. There have been many studies that investigated the sensitivities of the urban climate model's

input parameters and UHI mitigation scenarios. Though the previous studies did not use the PALM model, their contributions might be summarized and compared with this study. Unfortunately, I cannot find something new in this study conducted with the LES model. A clear scientific reason for conducting the sensitivity experiments using a PALM model needs to be presented first. Second, it is not clear why the two types of sensitivity experiments were organized. Besides, what is the reason for selecting the scenarios and model parameters used in the experiment? Before the sensitivity simulations, the reference simulation should be done with an optimized setting that can well simulate the actual (measured) meteorological conditions. In many studies, the sensitivity experiments presented in this study have been performed, and significant results have been presented. Actually, the results of this study are not beyond previous studies. Overall, this manuscript should be further improved by setting up more specific scientific questions and reanalyzing the sensitivity simulation results focusing on the scientific purpose.

Specific comments: L46-47: It should be focused on specific scientific problems. What do the 'systematic' sensitivity studies mean? L76-77: The boundary meteorological conditions might be critical in determining the meteorological conditions over the target domain, so more description of the boundary conditions will be helpful. How to feed the WRF output to the PALM model? What is the feeding time step? L87: Is there any reason to select PM10 for analysis rather than NOx, PM2.5? if vehicular emission is estimated, more relevant species might be NOx and PM2.5 rather than PM10. L90: More specific or technical description might be helpful for the mesoscale and microscale coupling strategy. How frequently is the WRF output provided? L93: Why didn't you use urban parameterization in the WRF simulation? Generally, the use of urban parameterization in WRF can give better simulations than the NoahLSM bulk urban parameterization. Providing realistic meteorological boundary conditions to the PALM model might be critical in the simulation over the target area. L103-105: which parameters were measured? What are the methods to get the parameters from the site? More specific descriptions will be necessary. L112-114: any reference for

Interactive
comment

Prague 3D model? L121-124: it seems that this spin-up result can influence the analyses of the sensitivity simulation results. L125: does the 'simulation' mean sensitivity or spin-up simulation? L134-135: Despite low vegetation fraction, this study says the vegetation has the most important factor in this area. Are there any studies to evaluate the physical parameterization of vegetation in the PALM model? If so, add the papers as references. L154-155: Please add a reference paper that explains the WRF and PALM coupling strategy. If this were done in this study, more description would be informative and useful. L172: why was the heatwave episode selected? I guess the series of sensitivity simulation results might depend on the case selection. L179: It seems that the PALM model does not cover the LLJ in the vertical direction. L187: Please check the emission unit in Fig. 3. It is difficult to read the emission intensity from the Fig. 3. What are the emission fluxes of NOx, PM2.5, and PM10 used in the simulation? How reliable are the estimated emissions? The primary pollutants emitted from passenger cars are NOx, CO, VOCs, and small particulate matter fractions. For PM10, blown-dust might be a major primary source on roads. L191-192: More explanation will be needed why the sensitivity tests are needed. What should be the base simulation in this study? What do you mean by 'real values'? How did you select the model parameters? L200-201: I think that the necessity does not worth publication. L212: Result section should be significantly revised. Tables 3 and 4 are too busy but show little differences. Many studies have reported similar sensitivity results. Please compare your results with them. Fig. 4: at which level are the variables plotted? Define the surface temperature. Please show how to calculate MRT and PET from the model results. PM concentration field looks unrealistic in magnitude and spatial distribution. Compare also Fig. 3. Is there any comparison against measurements?

---

## Referee Comment (RC2) · Anonymous Referee #2 · 14 Dec 2020

General comments: The sensitivity of the LES model framework PALM-4U to selected surface parameters and to a combination of surface properties representing potential planning scenarios are evaluated. While the study reveals some nice details on the capabilities of such LES modelling of a real urban environment, the study design could be clearly improved. - It is not clear why the scenarios are chosen in this particular way. For example, several planning scenarios appear rather unrealistic. - Further, scenarios discussed in full in the main manuscript should be reduced to only reveal the most important aspects of the modelling capabilities and urban planning assessment. Currently, it is very difficult for the reader to keep track of all the scenarios being discussed. This may also be related to the rather high number of figures which could be reduced to give a better overview to the reader. - A major drawback of the scenarios aiming to assess model sensitivity to certain input parameters is that no reference measurements are presented to determine model performance. So how do the SA scenarios really differ from the SB scenarios? - Finally, the analysis should really highlight the added value from the LES model setup, i.e. the spatial variability in meteorological indicators. Most of the "average" conclusions drawn (e.g. contrary impact on human thermal comfort and ventilation) can be expected from lower resolution modelling – there is a clear lack of references to e.g. evaluation of urban surface scheme.

Minor comments: P2, l50: PM10 defines particle up to diameter of 10 $\mu$m P3, l66: Why are two radiation models listed? What are they doing, respectively? P4, l87: How are the boundary conditions defined for the pollutants? Spatial variations in surface emissions? Horizontal advection and long-range transport? P4, l93: Why is there no urban scheme used in WRF? How does this impact the boundary conditions provided in the nesting (e.g. wind profiles, boundary layer height, . . .)? Are there relevant studies that should be cited here? P4, l108: so albedo and emissivity are independent of material category? P4, l110: define symbols at first occurrence P4, l150: Give some details on the building database. This a vector dataset? What is the level of detail? P5, l132: replace 'housing' by 'residential' or remove the word. P5, l144: what are 'non-impervious' anthropogenic surfaces? P6, Figure 1: include reference to Resler et al. (2017),in figure caption as this defines the 'old domain' P6, l151: provide reference for ABL height maximum in summer. P7, l160: why adding a flat buffer zone? How does this impact the flow? Provide a reference where readers can find more information on this aspect. P8, l177: where were the meteorological measurements conducted? Within the study area? What height above roof level? P8, l180: Maybe more appropriate to present results in local time rather than UTC? P8, l183: add reference for importance of traffic emissions. What fraction of PM10 in the area is from local traffic emissions? What is the role of other sources? What is the role of regional transport and local scale advection? P11, Table 2: some scenarios are rather unrealistic. Maybe provide some reasoning why these were tested? i.e. changing all roads to grass but then not changing vehicle emissions is a scenario that can not be translated to reality. P12, l222:

[Figure]

that's the reason why spin-up time is usually excluded from analysis. Please clarify you comment. P12, l228: please clarify. All indicators at 2m height above ground. What distance from the buildings? Which surface temperature is used as the indicator? P12, l229: what is meant by 'where necessary'? please explain. P13, Figure 4: Some odd model results should be discussed. This includes: very high LST in small gaps between buildings. Are these realistic? Where do they come from? Why do they not translate into high MRT? Also, Why are the PM10 concentrations only relevant in the two cross-roads? According to Figure 3 there are also emissions for the roads closer to the domain edges. Also, make sure symbols and variable names are defined and sued consistently. E.g. in the text and figure caption you use 'surface temperature' without defining the term 'LST' that appears in the Figure. P13, l154: Explain. Why is the importance of window fraction changing throughout the day? P13, l152: rephrase 'opposite behaviour to the air temperature in terms of the sign of the changes with higher absolute values'. Not clear. P18, l278: It is actually more interesting to see the spatial variability in impact of different scenarios. While average impact can be expected simply according to simple model physics and are in accordance with low-resolution simulations, the added value of the LES approach are the new insights into the spatial variability. This should be highlighted more clearly. But where are the analysis points marked that are shown e.g. in Figure 11? More detailed discussion could be nice. P19, Figure9: combine with Figure 10 to reduce the number of figures and make analysis more compact. P21, l330: Provide some interpretation. What explains the decrease in particle concentrations?

―――――――――――――――――――

---

## Author Comment (AC1) · 14 Feb 2021

**General comments** *This manuscript presents a series of sensitivity simulations of the PALM model in a real urban area in Prague, Czech Republic. The sensitive tests were conducted two-fold by changing the model's physical properties and changing urban morphological characteristics. All the simulations were conducted for a specific hot day over a small block area, and then their results were compared in terms of the influence of UHI and air quality. This study presents a potential capability of using the PALM model as a tool for urban climate research, and it seems that intensive computational works have been done for the sensitivity experiments. However, this study has some significant drawbacks that should be improved. First, it is not clear what this study contributes because its purpose is not clearly presented in a scientific sense. There*

*have been many studies that investigated the sensitivities of the urban climate model's input parameters and UHI mitigation scenarios. Though the previous studies did not use the PALM model, their contributions might be summarized and compared with this study. Unfortunately, I cannot find something new in this study conducted with the LES model. A clear scientific reason for conducting the sensitivity experiments using a PALM model needs to be presented first. Second, it is not clear why the two types of sensitivity experiments were organized. Besides, what is the reason for selecting the scenarios and model parameters used in the experiment? Before the sensitivity simulations, the reference simulation should be done with an optimized setting that can well simulate the actual (measured) meteorological conditions. In many studies, the sensitivity experiments presented in this study have been performed, and significant results have been presented. Actually, the results of this study are not beyond previous studies. Overall, this manuscript should be further improved by setting up more specific scientific questions and reanalyzing the sensitivity simulation results focusing on the scientific purpose.*

First of all, we would like to thank the reviewer for many interesting suggestions. Carefully, we went through all the comments and hopefully present a much-improved version of the manuscript.

Regarding the state-of-the-art, we agree with the reviewer that many existing studies have been performed, especially for what we call B-type scenarios, however, not many have been presented in this systematic manner. By that, we mean studies that would systematically test a larger set of parameters in the models. The studies that we are aware of usually dealt with one or two of these parameters (typically albedo and emissivity in the "synthetic" scenarios).

In fact, when we were preparing the simulations, we found out that setting up the model with real-life parameter values was a very complicated task. Either they are described in engineering tables in a certain range or only estimated. Furthermore, assigning one value to a specific surface itself brings a level of uncertainty. Our motivation (for the

[Figure]

A-type scenarios) was then to assess how much this uncertainty in input parameters can influence model results. We haven't found a single study using an LES model (or a climate model with an urban parameterization for that matter) that would test such an extensive set of parameters, therefore we are confident that our study brings quite a significant amount of new findings.

The reason we limit the comparison to LES models is mainly that simplified radiation models or climate models employing urban parameterizations do not give the same answers as a very local CFD model (see e.g. a discussion in a recent metastudy by Krayenhoff et al, 2021). Even at a very high resolution of a couple of 100 meters, the urban parameterization is based on a simplified representation of cities typically by a street canyon model with only a schematic representation of real conditions. Radiation models, on the other side of the spectrum, are useful for local temperature assessment, but only allow limited inclusion of the street-scale flow characteristics typically for calculations of thermal comfort, but are not useful for air quality studies, for which the dynamical part is essential. In that regard, we feel that from a model development point of view, comparisons need to be made with the same group of models. Of course, this does not limit the use of various model types in real-life applications. In fact, this analysis was performed within the framework of a larger project URBI PRAGENSI, in which mesoscale models were used alongside PALM, but each in their respective field of expertise.

We reformulated the introduction of the study to better show the motivation for the analysis. We also added some more citations of several recent papers and two comprehensive reviews to the state-of-the-art section and discussion.

Regarding the evaluation of the basecase simulation, we need to stress that in our case the focus is on the model sensitivity in realistic conditions, not necessarily real conditions. This semi-real setup allows to perform sensitivity tests in better controlled conditions then a fully real setup while it still provides realistic conditions. Preliminary tests, however, were performed and the simulation showed to capture realistic

Interactive
comment

conditions. A thorough evaluation is clearly beyond the scope of this manuscript, however, validation of the previous model version in the same area was performed in Resler et al. (2017) and the latest version has been evaluated (albeit in a different area) and described in another manuscript, also under revision for this special issue (https://doi.org/10.5194/gmd-2020-175).

**Specific comments** *L46-47: It should be focused on specific scientific problems. What do the 'systematic' sensitivity studies mean?*

Reformulated.

*L76-77: The boundary meteorological conditions might be critical in determining the meteorological conditions over the target domain, so more description of the boundary conditions will be helpful. How to feed the WRF output to the PALM model? What is the feeding time step? L90: More specific or technical description might be helpful for the mesoscale and microscale coupling strategy. How frequently is the WRF output provided? L154-155: Please add a reference paper that explains the WRF and PALM coupling strategy. If this were done in this study, more description would be informative and useful.*

Section 2.2.2 (WRF configuration) was extended with brief information about the offline coupling procedure. Detailed procedure is beyond the scope of this manuscript, and has not been published yet (detailed information about the principle of the PALM mesoscale coupling was published in Kadasch et al., 2020 (https://gmd.copernicus.org/preprints/gmd-2020-285/, the description of the details of WRF processing into PALM inputs was described in Resler et.al. 2020 https://doi.org/10.5194/gmd-2020-175), our software used for this processing is part of the PALM SVN distribution under the UTIL/WRF_interface directory together with a brief description of its utilization. We have added these references to the manuscript text.

The 1-hourly 3-D fields from WRF outputs (T, Q, U/V/W) were horizontally and vertically

interpolated (in that order) to the PALM model grid. Because the PALM model used a higher-resolution terrain that would differ from the coarse terrain in WRF by as much as tens of meters, the vertical interpolation had to include stretching of the atmospheric columns.

At the bottom, the atmospheric columns were shifted to match the PALM terrain, therefore there were no missing data below the original terrain and the surface effects from WRF were preserved. However, at higher altitudes, the atmospheric columns could not be shifted by the same amount, as that would introduce unrealistic horizontal gradients mimicking the terrain shift below. In order to avoid this, the atmospheric columns were stretched heterogeneously. The WRF model uses either sigma or hybrid vertical coordinates, our simulations use the hybrid option where the lowest level is terrain-following and the highest level is isobaric. For each column, the geopotential height of each level in the WRF data was recalculated using the same formula and parameters used in WRF for calculating the heights of the hybrid levels, however with the surface pressure altered to match the PALM terrain. The recalculated level heights were then used for linear vertical interpolation into the PALM Cartesian vertical coordinate system.

The interpolated 3-D fields were used as initial conditions for the first timestep and their top and lateral boundaries were used as boundary conditions for all timesteps. For the velocity fields, the total volumetric flux disbalance was calculated for each timestep as a sum of the volumetric inflow minus outflow for all boundaries. This residual volumetric flux was then divided by the total area of the five boundaries and subtracted from the respective inwards-directed velocity component for each boundary in order to make the inflow and outflow perfectly balanced, as is required by the incompressible equations used in PALM.

*L87: Is there any reason to select PM10 for analysis rather than NOx, PM2.5? if vehicular emission is estimated, more relevant species might be NOx and PM2.5 rather than PM10.*

We acknowledge this suggestion and for the revised version we chose PM2.5 for the analysis rather than PM10. On the other hand, considering the methodology of emission calculation the emissions among different species differ only by a multiplicative constant. As there is no chemistry and pollutants are treated as passive tracers the results (in a relative sense) would be the same for any of the three pollutants. We extended the emission description paragraph (2.6) to clarify this.

*L93: Why didn't you use urban parameterization in the WRF simulation? Generally, the use of urban parameterization in WRF can give better simulations than the NoahLSM bulk urban parameterization. Providing realistic meteorological boundary conditions to the PALM model might be critical in the simulation over the target area.*

This analysis is focusing on the model sensitivity in realistic conditions, not necessarily real conditions. Additionally, for this particular case study, no detailed measurements are available against which the WRF model simulation could be perfected for the particular location. The reason for using this case study is to stay comparable to our previous study (Resler et al., 2017).

Another manuscript, also under revision for this special issue (https://doi.org/10.5194/gmd-2020-175), describes a validation experiment with similar settings. The second study also discusses the reasoning why no urban parameterization was used. Briefly, in our extensive preliminary testing with the local URBI PRAGENSI project (http://www.urbipragensi.cz - only in Czech), the WRF quite surprisingly performed better without urban parameterization in comparison to background synoptic stations in Prague and mainly in terms of the temperature vertical profiles which are fundamental for providing boundary conditions to PALM (in contrast to surface values).

*L103-105: which parameters were measured? What are the methods to get the parameters from the site? More specific descriptions will be necessary.*

Parameters were collected in a specialized data collection campaign carried out in the

framework of the URBAN-ADAPT project (financed by the Technology Agency of the Czech Republic) by a project partner. Details can be found in the Resler et al (2017) paper (https://doi.org/10.5194/gmd-10-3635-2017) as referenced in the manuscript.

*L112-114: any reference for Prague 3D model?*

Added reference to Prague OpenData portal (description only available in Czech language) and basic information about the dataset.

*L121-124: it seems that this spin-up result can influence the analyses of the sensitivity simulation results.*

The spin-up period itself is not included in the analysis. Some influence may persist in the first hours of the "real" simulation after the spin-up. Given the enormous computational costs, it was not feasible to extend the simulation for another day. However, we have performed preliminary tests which showed that a longer simulation time does not change significantly. Moreover, as the results show, for most of the scenarios, the differences between respective simulations show up mostly after the sunrise, which gives the model enough time. Therefore, we think this approach is correct.

Added some clarification in the text.

*L125: does the 'simulation' mean sensitivity or spin-up simulation?*

By simulation we refer to the complete simulation, starting with the spin-up part and then continuing with the full LES run.

*L134-135: Despite low vegetation fraction, this study says the vegetation has the most important factor in this area. Are there any studies to evaluate the physical parameterization of vegetation in the PALM model? If so, add the papers as references.*

The paper describing land-surface modeling in the PALM model is currently also under review for GMD (https://doi.org/10.5194/gmd-2020-197, added to the model introduction section).

*L172: why was the heatwave episode selected? I guess the series of sensitivity simu-
lation results might depend on the case selection.*

Certainly, the results would be different. The reason for choosing specifically a heat-
wave episode was to stay in-line with the typical application of UHI mitigation studies
(as represented by the B-type scenarios in the study: increase of high reflective sur-
faces, tree shading, following thermal comfort etc). Additionally, the simulations per-
formed in 2018 and 2019 were, and frankly still are, so computationally expensive,
that repeating this exercise for more different cases (e.g. for winter) was technically
unfeasible within the time frame of the URBI PRAGENSI project, in which they were
performed.

*L179: It seems that the PALM model does not cover the LLJ in the vertical direction.*

The parent domain, with a vertical extent of 2.5 km, covers the LLJ at 640 m.

*L187: Please check the emission unit in Fig. 3. It is difficult to read the emission
intensity from the Fig. 3. What are the emission fluxes of NOx, PM2.5, and PM10
used in the simulation? How reliable are the estimated emissions? The primary pollu-
tants emitted from passenger cars are NOx, CO, VOCs, and small particulate matter
fractions. For PM10, blown-dust might be a major primary source on roads.*

We thank the reviewer for pointing out the discrepancies in emission fluxes in Fig. 3,
we corrected the values in the Figure, and in accordance with the other comment we
changed the displayed pollutant to PM2.5. We also added information about the ranges
of emission fluxes of the other two pollutants to the text.

Regarding the question of the reliability of the emissions. It is hard to estimate the
uncertainty of the emissions as the uncertainty of some input parameters are unknown.
One parameter is traffic intensity which is based on annual census data. Although
the exact vehicle numbers for modelled days are unknown we think this can serve as
a good estimate of the typical traffic load in the locality for the given period. Second

possible source of error is the assumption that only passenger cars are present. Based on the traffic census data there are only 2

*L191-192: More explanation will be needed why the sensitivity tests are needed. What should be the base simulation in this study? What do you mean by 'real values'? How did you select the model parameters?*

All simulations use the same dynamical inputs, the baseline differs only by using real values (ie. as measured or estimated based on the actual materials used in the real city), while the scenario simulations change them. We tried to clarify that better in the manuscript text. The sensitivity tests are described in the following two sections in detail together with the reasoning for why they're needed and how they were selected (A-type scenarios = typically material constants = assessment of the uncertainty in model outputs due to uncertainty in model inputs; B-type scenarios = typical UHI adaptation measures).

*L200-201: I think that the necessity does not worth publication.*

Reformulated

*L212: Result section should be significantly revised. Tables 3 and 4 are too busy but show little differences. Many studies have reported similar sensitivity results. Please compare your results with them. Fig. 4: at which level are the variables plotted? Define the surface temperature. Please show how to calculate MRT and PET from the model results. PM concentration field looks unrealistic in magnitude and spatial distribution. Compare also Fig. 3. Is there any comparison against measurements?*

We agree that the tables were a bit messy, a color code would certainly help, but it's not possible to have colors in the GMD LaTeX tables. Since both tables were in the supplement (TableS02) together with the rest of the variables and locations, we decided to eliminate these two tables from the manuscript and only reference the supplement. We added some more references to existing studies in the discussion, mainly two large

meta-studies. We also added a reference to papers by Frohlich and Matzarakis (2020) and Krč et al. (2020) describing MRT and PET and their implementation in PALM.

Regarding the PM10 concentration field, as the reviewer correctly pointed out, there were some discrepancies in emission fluxes (see also the answer for L187), for the revised version, we used PM2.5.
* * *

---

## Author Comment (AC2) · 14 Feb 2021

**General comments**

*The sensitivity of the LES model framework PALM-4U to selected surface parameters and to a combination of surface properties representing potential planning scenarios are evaluated. While the study reveals some nice details on the capabilities of such LES modelling of a real urban environment, the study design could be clearly improved. - It is not clear why the scenarios are chosen in this particular way. For example, several planning scenarios appear rather unrealistic. - Further, scenarios discussed in full in the main manuscript should be reduced to only reveal the most important aspects of the modelling capabilities and urban planning assessment. Currently, it is*

[Figure]

*very difficult for the reader to keep track of all the scenarios being discussed. This may also be related to the rather high number of figures which could be reduced to give a better overview to the reader. - A major drawback of the scenarios aiming to assess model sensitivity to certain input parameters is that no reference measurements are presented to determine model performance. So how do the SA scenarios really differ from the SB scenarios? - Finally, the analysis should really highlight the added value from the LES model setup, i.e. the spatial variability in meteorological indicators. Most of the "average" conclusions drawn (e.g. contrary impact on human thermal comfort and ventilation) can be expected from lower resolution modelling – there is a clear lack of references to e.g. evaluation of urban surface scheme.*

We would like to use this opportunity to thank the reviewer for many helpful suggestions. We added more discussion on the issues raised by the reviewer(s) throughout the manuscript and believe that the revised manuscript version is considerably improved and answers all the reviewers' comments. Some text was reformulated to be more comprehensible by the readers, namely, we attempted to make it more clear, why the two scenario types were employed, how they differ and what kind of questions they are designed to answer. We also added more discussion of the "non-average" model response, bearing in mind the need to keep the manuscript as concise as possible, we tried to find a compromise between highlighting the interesting results and not overwhelming the reader with descriptions of every single analysis point.

The present manuscript tries to really focus on the sensitivity analysis, for systematic validation against measurements we refer the readers to the accompanying manuscript under review for the same special issue (https://doi.org/10.5194/gmd-2020-175).

**Minor comments**

*P2, l50: PM10 defines particle up to diameter of 10 $\mu$m P3,*

corrected

*l66: Why are two radiation models listed? What are they doing, respectively?*

The Rapid Radiative Transfer Model for Global models (RRTMG, see Clough et al., 2005) is a one-column radiation model which calculates the global downward incoming radiation. The radiative transfer model (RTM, see Krč et al., 2020) takes these global radiation values and calculates explicit interactions of shortwave and longwave radiation with urban surfaces like terrain and buildings (shading, absorption, reflection, emission) and plant canopy (partial shading, absorption, emission). It provides temporally and spatially resolved radiative fluxes for the land and building surface models and plant canopy model for calculation of the surface and ground heat fluxes and plant canopy sensible and latent heat fluxes. The RTM also calculates detailed spatial and temporal structure of the mean radiant temperature (MRT) and provides it to the biometeorological module. We updated the formulation to better distinguish these two models.

*P4, l87: How are the boundary conditions defined for the pollutants? Spatial variations in surface emissions? Horizontal advection and long-range transport?*

The boundary conditions for pollutants are set to zero. We decided to study air pollution sensitivities to pure transport without chemistry as this makes these sensitivities easier attributable and thus more useful. As we study the sensitivity (differences of scenarios) and we concentrate only to traqnsport of the pollutant, the boundary conditions do not influence these differences. Note that we use boundary conditions from CAMx in the validation study GMD-2020-175 (currently under review), as well as in practical urbanistic studies. Emission is distributed spatially to particular streets and transport flow lines, crossroads,... according to the traffic census. Horizontal advection is carried by the PALM dynamic and chemistry module. The long-range transport, as much as it is important for weather and air quality forecasts, does not influence the sensitivity (differences) as far as it is considered the same in all scenarios.

*P4, l93: Why is there no urban scheme used in WRF? How does this impact the*

*boundary conditions provided in the nesting (e.g. wind profiles, boundary layer height, . . .)? Are there relevant studies that should be cited here?*

This analysis is focusing on the model sensitivity in realistic conditions, not necessarily real conditions. Additionally, for this particular case study, no detailed measurements are available against which the WRF model simulation could be perfected for the particular location. The reason for using this case study is to stay comparable to our previous study (Resler et al., 2017).

Another manuscript, also under revision for this special issue (https://doi.org/10.5194/gmd-2020-175), describes a validation experiment with similar settings. It also discusses the reasoning why no urban parameterization was used. Briefly, in our extensive preliminary testing within the local URBI PRAGENSI project (http://www.urbipragensi.cz - only in Czech), the WRF quite surprisingly performed better without urban parameterization in comparison to background synoptic stations in Prague and mainly in terms of the temperature vertical profiles which are fundamental for providing boundary conditions to PALM (in contrast to surface values).

*P4, l108: so albedo and emissivity are independent of material category?*

reformulated

*P4, l110: define symbols at first occurrence*

Added definitions

*P4, l150: Give some details on the building database. This a vector dataset? What is the level of detail?*

Added reference to Prague OpenData portal (description only available in Czech language) and basic information about the dataset.

*P5, l132: replace 'housing' by 'residential' or remove the word.*

corrected

*P5, l144: what are 'non-impervious' anthropogenic surfaces?*

simplified

*P6, Figure 1: include reference to Resler et al. (2017),in figure caption as this defines the 'old domain'*

Reference added

*P6, l151: provide reference for ABL height maximum in summer.*

Added references with a more accurate estimate of the ABL height in Europe in summer.

*P7, l160: why adding a flat buffer zone? How does this impact the flow? Provide a reference where readers can find more information on this aspect.*

With the mesoscale nesting we prescribe boundary value at the lateral and top model boundaries inferred from a mesoscale model, which is WRF in this case. However, since the modelled WRF flow does not necessarily reflect the flow in the LES, adjustment effects occur. These include, for example, an acceleration/deceleration of the flow behind the inflow boundary as well as on the outflow boundary to maintain mass conservation in the model caused e.g. by different effective roughness. The flat buffer zones in this setup aim to allow for undisturbed adjustment to give the model more flexibility. Here, we refer to Kadasch et al. (2020) where these effects are discussed in detail. The exact size of the buffer zones were chosen according to prior numerical experiments as well as experience. Without adding such buffer zones the flow will be accelerated/decelerated even stronger since the mesoscale flow would immediately hit the buildings, which in turn would create strong up- and downwind areas. With adding buffer zones we aim to relax this effect, though we have to say that we do not fully get rid-off it.

*P8, l177: where were the meteorological measurements conducted? Within the study area? What height above roof level?*

Added station location. It is 4km away from the study area, slightly further away than another professional meteorological station (Praha-Klementinum), on the other hand, it is in an urbanistically similar part of the city.

*P8, l180: Maybe more appropriate to present results in local time rather than UTC?*

We opt to use UTC for compatibility with other studies. We provide information on local time in the text where appropriate.

*P8, l183: add reference for importance of traffic emissions. What fraction of PM10 in the area is from local traffic emissions? What is the role of other sources? What is the role of regional transport and local scale advection?*

We extended the emission description paragraph (2.6) with the information on the traffic emissions ratios in comparison with the total emissions. Based on calculations using the regional chemical transport model (not published) the regional background is 50-60

*P11, Table 2: some scenarios are rather unrealistic. Maybe provide some reasoning why these were tested? i.e. changing all roads to grass but then not changing vehicle emissions is a scenario that can not be translated to reality.*

This less realistic approach was adopted to separate the effects of UHI mitigation techniques that are at the forefront of urban planning now. Changing vehicle emissions too would make it difficult to distinguish the meteorological effects on air quality.

*P12, l222: that's the reason why spin-up time is usually excluded from analysis. Please clarify you comment.*

The spin-up period itself is not included in the analysis as it does not represent the real simulated conditions (the whole dynamics of the model is parameterized). The spin-up technique is used for bringing the temperature of the ground/wall layers closer to the reality and thus avoid the strong balancing in the first simulation hours which would significantly affect the simulation results. Some influence may still persist in the first hours of the "real" simulation after the spin-up. Given the enormous computational

costs, it was not feasible to extend the simulation for another day. However, as the results show, for most of the scenarios, the differences between respective simulations show up mostly after the sunrise, which gives the model enough time. Therefore, we think this approach is correct.

Added some clarification in the text.

*P12, l228: please clarify. All indicators at 2m height above ground. What distance from the buildings? Which surface temperature is used as the indicator?*

Surface temperature = material skin layer temperature (results of energy balance equation calculation).

The distance from the buildings is given by the respective measurement point (see the map), where point measurements are analyzed. The model itself does not calculate influence only of the one nearest surface but it considers and calculates the impact of all surfaces in the model on conditions at the given evaluation point.

*P12, l229: what is meant by 'where necessary'? please explain.*

Removed the sentence, the only other variable that was used was the net radiation for discussion of albedo scenarios.

*P13, Figure 4: Some odd model results should be discussed. This includes: very high LST in small gaps between buildings. Are these realistic? Where do they come from? Why do they not translate into high MRT? Also, Why are the PM10 concentrations only relevant in the two cross-roads? According to Figure 3 there are also emissions for the roads closer to the domain edges. Also, make sure symbols and variable names are defined and sued consistently. E.g. in the text and figure caption you use 'surface temperature' without defining the term 'LST' that appears in the Figure.*

Ad LST) We agree with the reviewer that these striking high land-surface temperatures should be discussed in the text (the issue was alluded to in the model limitations section in the discussion of pollutant concentrations, temperature was added in the revised

version). Indeed, this is a well-known issue for the PALM developers that within narrow structures quite high values of air and surface temperature can occur. This is a numerical artefact and is attributed to the topography formulation in PALM in conjunction with the nature of LES and the numerics. Within such narrow geometries, the flow is not well resolved on the numerical grid and largely affected by numerical dispersion errors, forming stationary numerical oscillations within such geometries. As the turbulent mixing is almost not resolved there, no turbulent eddy can penetrate into these geometries, so that almost no mixing with the air above happens. Geometries that are represented by only one grid point are filtered in the initialization, but larger (though still narrow) geometries remain. We do not observe such high temperatures all the time, only at some times and also not within each narrow geometry.

Here, we also note that this is only an issue for the prognostic variables that are solved on the numeric grid. The MRT, however, is a diagnostic quantity that results from radiation reflections from various horizontal and vertical walls and thus the effect of high surface temperatures at single surface partly cancels out.

Ad PM) Regarding the PM concentrations (formerly PM10, changed to PM2.5 in the new version, see other comments), the original Figure 3 confusingly showed all emissions in the domain and the closest streets. However, the model inputs included only emissions from the two crossing streets (Dělnická and Komunardů). We supplied a new version of Figure 3 with only those emissions actually included in the inputs.

*P13, l154: Explain. Why is the importance of window fraction changing throughout the day?*

Our hypothesis, supported by the daily cycle (chaotic behaviour during the high sun period of the day and more systematic changes in the late afternoon and through the night) is that the reason for this lies in the different heat storage by the windows and the wall. Simply put: lower window fraction = more walls = more heat storage = higher temperature and vice versa. We extended the discussion in the text accordingly.

*P13, l152: rephrase 'opposite behaviour to the air temperature in terms of the sign of the changes with higher absolute values'. Not clear.*

Rephrased, an example is shown in the following sentences.

*P18, l278: It is actually more interesting to see the spatial variability in impact of different scenarios. While average impact can be expected simply according to simple model physics and are in accordance with low-resolution simulations, the added value of the LES approach are the new insights into the spatial variability. This should be highlighted more clearly. But where are the analysis points marked that are shown e.g. in Figure 11? More detailed discussion could be nice.*

All points and averaging areas are shown in the supplement figures S01-S09. We extended the discussion of the "non-average" model response by adding more examples of the spatial variability at the end of this section, in the discussion of the influence on concentrations and also in the Discussion.

*P19, Figure9: combine with Figure 10 to reduce the number of figures and make analysis more compact.*

Accepted.

*P21, l330: Provide some interpretation. What explains the decrease in particle concentrations?*

The decrease of concentrations in the Komunardů street (see maps in the supplement) is in fact only concentrated in and in the vicinity of the crossroad, no significant changes are evident further away from the crossroad. The maps of the wind speed can suggest an explanation. The wind over the roofs runs approximately from the east. In the basecase, this causes the air flow is running almost freely through the Delnicka street (west-east oriented street) and it creates quite a nicely pronounced anticlockwise eddy in the Komunardu street (north-south street) which is broken in the area of the crossroad by the east-west stream from Delnicka street. Planting a tree alley only in

the Delnicka street (scenario SB13 and SB14) has two consequences. The first is that the air flow in Delnicka street is slowed down which allows the manifestation of the air buoyancy near the south-facing wall caused by radiative heating of this wall during the day. This causes an increase of the concentrations in the north half of the street and a modest decrease of them in the south parts of the street canyon. The second consequence affects the area of the crossroad. The obstruction in the Delnicka street has the effect that the area of the crossroad behaves as a part of the north-south street canyon which leads to partial creation of the eddy similarly to other parts of the Komunardu street. This increases the transport of the crossroad emission from transportation to the space above the roofs and leads to decrease of the concentrations in the crossroad area. While the statistics for particular streets are done over all their surface including the crossroad area, the average concentration in Komunardu street decreases. This situation also suggests how complex the relations of the quantities in the urban canopy are and how difficult it is to find any generalized rules of them. We added a shorter version of this rather long explanation in the text.

---

## Author Response (AR2)

**Topical Editor Decision: Reconsider after major revisions** (29 Mar 2021) by for clarification: the reviewer of the second round was reviewer #2 of the first round.

**I went over the complete manuscript today.**
1.) I appreciate the systematic nature of sensitivity tests.

Thank you for the thorough review and many helpful suggestions. The systematic approach was in fact our main goal here. As commented on in various places, the aim was to provide an extensive sensitivity study and therefore some aspects of general model evaluation are not discussed here. For that we refer to either a previous paper by Resler et al. (2017) or a companion paper by Resler et al. (2020) also in review for this special issue.

2.) I miss a discussion on how general your results are. Do you expect similar results for other neighborhoods?Which of the observed phenomena in your specific test case would expect to occur in other cases as well.

We added discussion in the first paragraph of the Conclusions section. In brief, the transferability to other locations largely depends on the type of analysis required. Neighborhood-wide response will most likely be comparable, however, when analysing very local influences, the sensitivity may vary considerably, not only in values, but at places even in the sign of the response.

3.) In the model limitation section you talk about missing model components (like variable building temperatures). In my opinion, the model system already includes many processes. What is your general strategy in demonstrating that the model produces plausible results.

We realize that calling the section "Model limitations" is a bit misleading, as some components are present in the model but not used for the study (discussed for each point, most of them can be neglected in the study settings). We renamed the subsection Study limitations and changed the formulation a bit (we also reorganized the Discussion and conclusions section to have a more logical structure).
We added a paragraph at the end of the limitations section clarifying the issue of validation, which is beyond the scope of this manuscript, however, extensive validation has been performed in another study by Resler et al. (2020) currently also under revision for this special issue to which we refer the readers.

**About the presentation style:**
My biggest problem is that you show many plots, but they are often not explained in the text. They are often only referenced, but not explained in detail.
This also implies that it is not made clear what you want to demonstrate with each figure.
This gives the reader a bit the impression that the figures are randomly selected.

We went thoroughly through the text and reorganized many of the figures to better capture the message of the paper (see below for individual details) and added discussion where appropriate. Among other changes (described below as replies to specific comments) we:

- Joined Fig.1 and 2 into one
- Moved Fig.3 into the supplement
- Moved Fig.10 into the supplement
- Joined Figs. 11 and 12

Would it make sense to create some summary plots where the sensitivity of a few selected variables (like average temperature difference) is shown for all scenarios. Fig. 17 in doi:10.5194/acp-14-2713-2014 hopefully makes clearer what I mean and can serve as an inspiration.

We added a summary boxplot for air temperature (Fig. 3) in the beginning of the Results section (added other variables in the supplement in order not to make the manuscript even longer).

**Minor comments (copied from the annotated PDF)**
L. 20: redundant stop
Corrected

L. 43: missing references to SOLWEIG and RayMan models
Added

L. 102&104: missing Sect. in cross reference
Corrected

L. 120: window model not mentioned in previous text
Reformulated (window treatment in the model is a part of the Building Surface Model - BSM, which is described in previous text).

L. 127-130: What is the motivation behind using two overlapping runs? Do mean partially overlap or do they simulate the identical period? How are the two horizontal resolutions applied?
Reformulated.

Boundary conditions for PALM are generated from the inner 3km WRF domain, the 9km outer domain serves as a standard way to deal with a large resolution jump from the global analysis to the regional model (double-nesting).

L. 135: misplaced parentheses in citation
Corrected

L. 219: missing citation

Corrected

L. 219: stop instead of comma
    Corrected

L. 227: In the results section, you use PM2.5 as a proxy for air quality. But without chemical modelling and treating only the dispersion as a passive tracer, I am not sure if a link to air quality should be made. At least it should be stressed in the text, that only dispersion of the pollutant is considered.
    We added a short clarification of this and referred the reader to the complete discussion of this issue in the "Limitations" section.
Fig 3.: What does EPSG: 32633 mean?
    Clarified (Projection: WGS 84/UTM zone 33N)

L. 245: is SA used for the baseline simulation or the 21 sensitivity studies? I assume for the latter. Then move the text inside the brackets one line up.
    Corrected

L. 288: Why do you show this figure as it treats a rather unimportant parameter variation? Why not choose instead to show the differences in T and MRT for SA1,SA2 in a style similar to Fig. 5.
    Moved the figure to the section discussing surface temperature. Added more discussion of the relevance of the figure (briefly: it may not have a huge influence on average, but is important locally in green areas or their vicinity).

L. 294: at around
    Corrected

L. 301: not clear what "higher absolute values" refers to
    Reformulated

L. 307: It is not clear if this holds for all SA scenarios or only SA1 and SA", which were discussed in the subsequent paragraph.
    Corrected

Fig.6: What is the information gain of the second panel?. Are there any characteristic differences between those two panels that are explained in the text? If not, you could remove one panel.
    The main point here is that the responses are highly heterogeneous and non-symmetric for symmetric scenarios, mainly in the spatial distribution. We added some discussion on that to the text.

L. 324: Do not leave it to the reader to find out which row of Fig.9 shows which case. Again if you show four panels, then a minimum amount of description is needed in the text.
    Added description to text.

L. 329: This figure shows 6 different lines. Again, the description in the text is cursory.

Added description to text, figure moved to supplement.

L. 330: If you increase resolution, variables are allowed to vary on smaller scales. Thats agreed on. But what is the added value? Are those more refined patterns really better? Do you know from a comparison with observations, that the higher resolution really leads to an increased prediction skill?

We replaced the "added value" with "importance" here, as we understand that talking about added value is misleading as it has quite a firmly established meaning to which we cannot attest in this study. The design of this experiment was semi-synthetic and was not intended for performance evaluation and comparison of the results to observations. For validation experiments, we refer to our previous validation study (Resler et al., 2017) and the one in review for this special issue (Resler et al., 2020).

Fig. 7: If you intend to show this figure in the manuscript, its content should be explained with more than just a few words.

More discussion was added in the text for this figure (now Fig. 5) and the relevant Fig. 4. We also replotted the figure to show differences of the scenario to basecase which allowed the figure to be shorter yet (hopefully) more intelligible.

Fig. 8: the y axis label says PM10 not PM2.5
Moreover, the caption does not explain the solid and dotted line styles.
Again: If you intend to show this figure in the manuscript, its content should be explained with more than just a few words. I can only see random time series and do not understand what you want to convey with this figure. You have to explain it in the text.

(Now Fig.6) We understand that the choice of the figure was not ideal, we replaced it with a better example of spatial and temporal heterogeneity of the response in the albedo changing scenarios. Also we added a description in the text.

Fig. 9: it would be better include this information (*point designations*) as a legend in each panel. Otherwise you have to scroll down. Or use a 2x2 placement of panels.

(Now Fig.7) Added point designations to the panels. We discussed using the 2x2 panel, but unless the image was rotated, the 2x2 placement would lose a lot of detail.

L. 455: Is it model limitation or a limitation of the study design?

That's a great point, we changed the section to "Study limitations" to cover both aspects, changed the formulation slightly in the intro and added a paragraph at the end of this subsection discussing the plausibility of the results given all the limitations mentioned.

L. 460: That's the kind of discussion that I missed and tried express in a previous comment.

We added the short mention in the previously mentioned paragraph and referred the reader to this discussion as we feel it is better to have the detailed discussion in this section than disrupt the flow of the text in the experiment setup section.

L. 483: Unnecessary appendix
Removed

**Second round reviewer's comments**

Although some improvements have been made to the mansucript, I would further encourage the authors to
a) better explain the rationale for selecting the scenarios analysed

We clarified the motivation behind the scenario selection for both types of scenarios, although we feel that this comment was perhaps more aimed at the B-type scenarios (the A-type scenarios are simply selected to cover all important material and surface parameters present in the model, so the choice there was obvious). The B-type scenarios (renamed *urbanistic*) were designed together with urban planners from the City of Prague in the framework of the Urbi Pragensi project, not necessarily to be realistic in all cases, but to provide them with some sort of "envelope" response, or upper and lower limit of the possible developments in the neighborhood (both "in" and "contrary" to the direction of UHI mitigation). We added the clarification to the introduction of the sect. 2.7.2.

b) reduce the number of figures and make the analysis more accessible to the reader

- Joined Fig.1 and 2 into one
- Moved Fig.3 into the supplement
- Moved Fig.10 into the supplement
- Joined Figs. 11 and 12
- Added more discussion to the figures, that were perhaps a bit neglected in the previous version

c) better highlight the added value of LES modelling and put findings in relation to existing studies evaluation urban planning scenarios.

The added value, as far as the established meaning in the modeling community goes, cannot be really corroborated by our study as we didn't do any extensive evaluation of the model results against observations, except preliminary checking that the baseline simulation produces realistic results in line with the previous paper of Resler et al. (2017). However, in the discussion we added references to other studies that validated LES and RANS against wind-tunnel

measurements and all conclude that LES shows much better performance than RANS.

L323: not clear. What are "materials used in reality"?
    clarified

Section 2.7.1 and 2.72 – the description and rational for the design of scenarios is still not entirely clear. While "synthetic" scenarios are designed to understand the impact of parameter uncertainty, the real life scenarios are evaluating planning strategies. However, more details should be provided explaining why the specific scenarios are selected. For example, why would you test a scenario without any vegetation? Is this considered a "worst-case example"? Maybe the term "real life" scenario is not suitable if the setups are actually not all realistic such as when removing roads but not the cars.
    As mentioned previously in the reply to point *a)*, we discussed these scenarios with the Prague City authorities and the aim was not always to analyze super-realistic scenarios, but to provide some sort of upper/lower limit assessment (e.g. removing all trees, changing all surfaces to asphalt etc.). We agree that calling these "real-life" was a bit of a stretch, so we changed it to "Urbanistic scenarios".
    The other point about removing roads but not car emissions was in fact motivated by the fact that changing both at once could potentially mask the influence of vegetation on dynamical drivers on air quality, when the car emissions serve more as a tracer as we do not consider any chemical reactions.

---

## Author Response (AR3)

**Topical Editor Decision: Publish subject to minor revisions (review by editor)** (07 Jun 2021) by Simon Unterstrasser
Comments to the Author:
Dear authors,

I appreciate the latest text modifications.
Unfortunately, there is one thing that has to be fixed and which I probably overlooked in the beginning.
GMD is strict about reproducibility. It is required that you provide all scripts, code and data that allows other to reproduce your plots.
This includes configuration files of all performed simulations in addition to the source files given in https://data.uni-hannover.de/dataset/palm-6-0-r4093. In theory, anyone should be able to re-do your simulations.
Moreover, all presented plots must be reproducible.
Best practise is to bundle all plot scripts together with the necessary input data (i.e. the output of the PALM simulation that is used in the plot procedures) in a Zenodo data set.
See for example: https://data.uni-hannover.de/dataset/6687444f-a3a7-45b8-8453-a39995b8e4d4

Best wishes,
Simon

Dear Simon,

We bundled model configuration files, input data needed for running the simulations, model outputs postprocessing code, i.e. extraction and visualisation scripts, together with necessary data extracted from the raw model outputs together with the the additional outputs (mentioned but not discussed in the manuscript) to the permanent repository of scientific outputs at the Czech Academy of Sciences, as we discussed after the first submission (for which we only used it to store the additional outputs) at http://hdl.handle.net/11104/0309669. We changed the text in the "Code and data availability" section accordingly.

Best regards,

Michal Belda on behalf of the team.